# Intrinsic motivation in virtual assistant interaction for fostering spontaneous interactions

**Chang Li**◉*, **Hideyoshi Yanagisawa**◉*

Graduate School of Engineering, The University of Tokyo, Tokyo, Japan

* lichangtx@gmail.com (CL); hide@mech.t.u-tokyo.ac.jp (HY)

**Data Availability Statement:** All relevant data are within the paper and the Supporting Information files. The audio files used to conduct the experiment can be accessed from https://datadryad.org/stash/landing/show?id=doi%3A10.5061%2Fdryad.9cnp5hqgf.

## Abstract

With the growing utility of today's conversational virtual assistants, the importance of user motivation in human–artificial intelligence interactions is becoming more obvious. However, previous studies in this and related fields, such as human–computer interaction, scarcely discussed intrinsic motivation (the motivation to interact with the assistants for fun). Previous studies either treated motivation as an inseparable concept or focused on non-intrinsic motivation (the motivation to interact with the assistant for utilitarian purposes). The current study aims to cover intrinsic motivation by taking an affective engineering approach. A novel motivation model is proposed, in which intrinsic motivation is affected by two factors that derive from user interactions with virtual assistants: expectation of capability and uncertainty. Experiments in which these two factors are manipulated by making participants believe they are interacting with the smart speaker "Amazon Echo" are conducted. Intrinsic motivation is measured both by using questionnaires and by covertly monitoring a five-minute free-choice period in the experimenter's absence, during which the participants could decide for themselves whether to interact with the virtual assistants. Results of the first experiment showed that high expectation engenders more intrinsically motivated interaction compared with low expectation. However, the results did not support our hypothesis that expectation and uncertainty have an interaction effect on intrinsic motivation. We then revised our hypothetical model of action selection accordingly and conducted a verification experiment of the effects of uncertainty. Results of the verification experiment showed that reducing uncertainty encourages more interactions and causes the motivation behind these interactions to shift from non-intrinsic to intrinsic.

## 1 Introduction

Virtual assistants (also termed "voice assistants") in today's market serve users through conversation. The competition among Amazon Alexa, Google Assistant, and similar systems does not involve only their usefulness. Developers want to build likeable virtual characters, not mere autonomous toolboxes. Recent studies shifted to topics such as affective experience [1] and hedonic benefits [2] during the interactions. From the perspective of affective engineering, these

**Funding:** H. Yanagisawa received KAKEN grant number 18H03318 from Japan Society for the Promotion of Science (https://www.jsps.go.jp/english/). The funders had no role in study design, data collection and analysis, decision to publish, or preparation of the manuscript.

**Competing interests:** The authors have declared that no competing interests exist.

aspects of virtual assistants are inseparable from user motivation. However, user motivation, although generally recognized as an important factor in determining the attractiveness of contemporary and future virtual assistants [3], has been little studied in this context. Here, one must also distinguish between different types of motivation: intrinsic motivation is derived from the expectation of enjoyment through taking an action, whereas extrinsic motivation is oriented toward the consequences of the action [4]. Between extrinsic and intrinsic motivations, there is another category, epistemic motivation, derived from the need to gain knowledge or to hone skills, with the hope of benefiting from progress eventually [4]. In the field of human–artificial intelligence interaction and in related fields, such as human–computer interaction, previous studies have scarcely discussed intrinsic motivation. Those studies either treated motivation as an inseparable concept or focused on non-intrinsic motivation, that is, the motivation to comply with tasks (extrinsic) [5,6] or to gather information for pragmatic usages (extrinsic or epistemic) [1]. Although there were studies addressing concepts related to intrinsic motivation (such as interest and curiosity), the affecting factors of intrinsic motivation remain unstudied [7].

Intrinsic motivation should be the next focal issue in virtual assistant interaction because it is crucial for enhancing the perceived smartness and likeability of future smart products. In the fields of education and consumer psychology, intrinsic motivation has been found to be related to higher levels of enjoyment, performance, and immersion in an activity [4,8,9]. However, virtual assistants are typically treated as autonomous tools by adult users, not as potential friends. A study found that while children are willing to befriend virtual assistants found in household smart products, adults often are not [10].

The main objective of the present research is to obtain knowledge about intrinsic motivation and intrinsically motivated interactions in virtual assistant interactions. We wish to learn under what circumstances the user would interact with a virtual assistant "just for fun." Eventually, we wish to design the virtual assistant's interaction strategy in order to foster spontaneous interactions, thus improving product engagement.

In everyday life, users interact with virtual assistants of their free will. In contrast, in laboratory studies, there are cases where the experimenter assigns interactive tasks to the user (participant). In such tasking scenarios, the participant is typically told to interact; therefore, such interactions are not "based on free will". To distinguish these two cases, we introduced the concept of "spontaneous interaction" in our study and specifically in our experimental design. We defined a spontaneous interaction as any interaction that is taken by users of their own will, whereas if the user is told to act (or is told to achieve a goal and acts accordingly), the action is not spontaneous. The motivation behind spontaneous interactions is best discussed by relating action selection to motivation in the context of generalized user–product interaction. We have constructed a theoretical model of the process by which intrinsic motivation is formed. On the basis of this model, we have proposed and tested hypotheses concerning the relationship between motivation, expectation, and uncertainty.

In the following sections of Section 2, we first introduce how expectation affects the action selection process (Section 2.1). Then we discuss effects of expectation and uncertainty on intrinsic motivation in Section 2.2. In Sections 2.3 and 2.4, we propose hypotheses on effects of expectation and uncertainty on these two factors.

To verify the hypotheses, we conducted a two-by-two experiment, which is introduced in Section 3. The results of the experiment did not support our hypotheses (discussed in Sections 4 and 5), such that we partially revised our model and proposed another hypothesis on uncertainty. The verification experiment on the effects of uncertainty is introduced in Section 6. The results and discussion are presented in Sections 7 and 8. In the final section, Section 9, we summarize our findings on the mechanism of intrinsic motivation and propose recommendations for designing better virtual assistants.

## 1.1 Expectation-based action selection

In a simplified setting, the interaction process consists of four phases [11]: 1) expectation, in which the user estimates the consequences of the intended action; 2) action; 3) observation, in which the user discovers the actual consequences of the action; 4) learning, in which the user assesses the differences between expectation and observation.

Expectations can be updated by learning. When what is observed is unexpected, the user tends to re-explore the product and to gather information about what caused the unexpected result [11]. Each time the user learns about the product, the expectations are updated, which in turn may bring changes to future action selection strategies. Learning is an on-going process. In reality, in user–product interaction, the user can only interpret observations and create strategy accordingly but can never acquire all the information available in principle [11]. This is because the interface is not transparent, which inhibits the user from knowing what is happening inside the machine. Nevertheless, users can explore a product until they gather sufficient knowledge to establish habits that they retain. For instance, one can use a computer without necessarily knowing how the hardware works, owing to the graphic user interface.

Regarding user motivation in virtual assistant interactions, a crucial aspect is the expectation of the assistant's capability. The users might ask themselves: "How smart is it, after all?" or "Where is its limit?" Accordingly, when the user has an intention ("I need to book a flight"), expectation of interaction result is formed before asking. If the user holds an expectation (of the assistant's capability) lower than what the intention would require, they are unlikely to initiate an interaction for booking ("I'd rather book it from my smartphone app!"). In a recent survey-based research, many of the interviewees mentioned that sometimes the use of virtual assistants made reaching their goal slower [7]. The user motivation (to interact) is clearly vulnerable to low-capability expectation with respect to the difficulty of the intended tasks.

While the aforementioned situations involved only extrinsic motivation (pragmatic intentions), we argue that intrinsic motivation would be damaged equally. When the user's intention is to seek enjoyment or reduce boredom, which by definition is intrinsically motivated, such motivation can also be damaged if the capability expectation in terms of entertainment capability is low. Furthermore, in such a situation, the user just "won't bother" to interact for fun. For instance, the user would not choose to chat with the assistant to pass time, if they believed that the assistant was not smart enough to adhere to the topic.

Unlike video games, which serve solely entertainment purposes, virtual assistant carries both pragmatic and entertainment functions. It is likely that capability expectations of a virtual assistant are formed throughout daily usage, where expectations of pragmatic and entertainment capabilities are not independent of each other. In other words, both intrinsic and non-intrinsic motivation are affected by the capability expectation. This statement is based on a mapping between motivation type and interaction type. Interactions that are enjoyment-oriented are intrinsically motivated (by the definition of intrinsic motivation [4]). Interactions whose goal is to complete tasks or raise productivity are extrinsically motivated (by the definition of extrinsic motivation [4]), hence non-intrinsically motivated. Interactions whose goal is to gather information or to learn are driven by epistemic motivation, and therefore also non-intrinsically motivated [12].

We have established that motivation and action selection are affected by expectation; thus, the next question is: what factors affect the forming of expectations and the learning process?

## 1.2 Expectation and uncertainty in virtual assistant interactions

In a previous study addressing expectation-based sensory perception (of physical property such as weight, denoted by θ in Eq 1), a computational model was proposed and validated, explaining

how human perception (a posteriori distribution, $\hat{\theta}$) follows certain probability distributions. As shown in Eq 1, perception $\hat{\theta}$ is formed based on expectation (a priori distribution, P(θ)) and information gained through observations (the likelihood, P(R|θ) where R is the firing rate distribution of the sensory stimulus θ). The bias between the a priori and the a posteriori (the bias is termed "expectation effect") is a function of uncertainty, prediction error, and noise [13]. According to this model, the estimation follows the Bayesian estimator [13–15]:

$$\hat{\theta} = \frac{V[P(\theta)]E[P(R|\theta)] + V[P(R|\theta)E[P(\theta)]]}{V[P(\theta)] + V[P(R|\theta)]} \tag{1}$$

In Eq 1, E[P(θ)] and V[P(θ)] represent the mean and the variation of the distribution P(θ), respectively. Although this model was proposed to explain sensory perception, the knowledge is transferable to virtual assistant interactions. Here, we borrow Yanagisawa's model [13] to understand expectation and perception of the virtual assistant's capability. In Fig 1, the horizontal axis represents the assistant's capability. The observation mean indicates that the assistant is capable of tasks easier than the difficulty represented by the mean; meanwhile, it indicates that the assistant is incapable of tasks to the right of the mean difficulty. Similarly, the prior expectation mean indicates the expected capability before observation.

Following the Bayesian model [13], if the expectation variation and observation variation are the same, the perception of capability should be in the middle of expectation mean and observation mean. When two variations are unequal, the perception should always be closer to the mean with smaller variation. In the case illustrated by Fig 1, the prior expectation has smaller variation; therefore, the perception is closer to expectation.

The prior variation indicates the prediction uncertainty: in the Bayesian estimation, a small variation in prior expectation indicates small uncertainty, and a large variation in prior expectation indicates large uncertainty in the prediction. Visually, the larger the uncertainty, the flatter the distribution curve. Likewise, the observation variation indicates the noise during sensory perception.

The capability expectation is learned through previous interactions. The user compares the assistant's expected capability against the expected "task difficulty", i.e., the minimal capability required to cope with the user's intention. If the assistant fails "rather easy" tasks, which are expected to be within the capability, the observation will be on the left of the expectation mean; hence, the posterior perception of capability will shift to the left. ("The assistant is not as capable as I thought.") Conversely, if the user observes that the assistant copes with difficult tasks, which are expected to be beyond its capability, the posterior perception of capability will

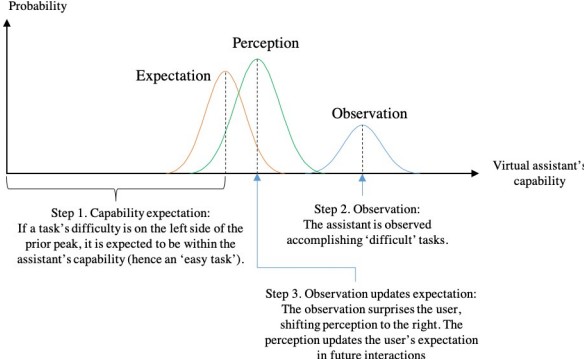

**Fig 1. Expectation and perception of the virtual assistant's capability.** Unexpected observations update the user's expectation of the assistant's capability.

shift to the right. ("The assistant is more capable than I thought.") Coping with tasks that are expected to be within the capability or failing tasks that are expected to be beyond the capability does not affect the mean of posterior.

In the long run, posterior perceptions update the user's experience, thus affecting the prior expectation in the future. In virtual assistant interaction, inconsistent performance by the virtual assistant gives rise to uncertainty. For instance, consider the following scenario: a virtual assistant complies with a task, and moments later, it fails to comply with an apparently similar task. Observing such inconsistent performance results in a flattened posterior distribution compared with when uncertainty is small. In other words, the user becomes uncertain about the virtual assistant's capability.

Thus, we can conclude that the virtual assistant's capability is perceived in consideration of "task difficulty" and that uncertainty weakens the confidence that the user places in their expectation. The relationship of expectation and uncertainty with intrinsic motivation is discussed in the following paragraphs, and a hypothesis of their effects on intrinsic motivation is proposed.

## 1.3 Hypothesis on expectation

From here on, we use "high" versus "low" to describe capability expectation: high expectation means the assistant is expected by the user to be able to cope with difficult tasks or satisfy complex intentions. Moreover, high versus low, as well as difficult versus easy, are discussed using relative measurement. Considering the capabilities of virtual assistants at present, a real-world example of an easy task would be "Set a 3-minute timer." Conversely, a difficult (if not impossible) task would be "Wake me up at 4 a.m. if Team A still has a chance to advance", when in reality, the chance is dependent on another game that ends at 3:30 a.m.

As discussed in Section 2.1, the user's motivation to interact with a virtual assistant is limited by its expected capability. The higher the expectation, the more rational it is for the user to believe that their intention will be satisfied through interaction. The lower the expectation, the more likely the user would "not bother" to interact. Therefore, we hypothesized that high expectation has positive effects on both intrinsic and extrinsic motivation to further interact with the assistant.

## 1.4 Hypothesis on uncertainty

As we discussed in Section 2.2, uncertainty can be perceived through observing inconsistent performances by the assistant. From here on, we use "small" versus "large" to describe uncertainty. The uncertainty is small (if not non-existent) when the assistant's performance is always consistent in response to the same type of task. Conversely, the uncertainty is perceived as large if the assistant copes with rather difficult tasks, and later fails at easier tasks of the same type. Similar to expectation, large versus small uncertainty is discussed using relative measurement. Another way to interpret uncertainty is to ask: "How capable is the assistant, after all?" The more difficult to judge its capability, the flatter the prior distribution (as a result of larger uncertainty). In our experiment, to introduce large uncertainty we deliberately made the virtual assistant perform in an inconsistent manner.

We hypothesized that the effect of uncertainty on motivation depends on the level of expectation. If the user holds high expectation of an assistant, the intrinsic motivation to further interact will be greater if small uncertainty has been perceived during previous interactions. If the expectation is low, we hypothesized that uncertainty would work in the opposite direction.

Fig 2 illustrates the reasoning behind the hypothesis. Assume that the "task difficulty" of an intention is indicated by the vertical dashed line. When the expectation is low, the prior mean

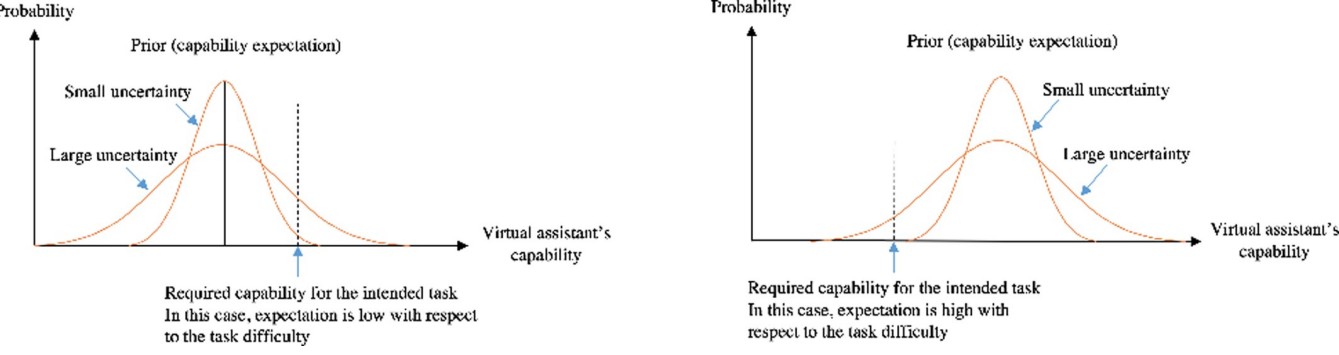

**Fig 2.** a. Hypothesis on uncertainty. A supposedly difficult task "intersects" with the large uncertainty prior distribution at higher probability than when uncertainty is small. When the expectation is low, it is more rational to expect an assistant exhibiting large uncertainty to perform better than it usually does. b. Hypothesis on uncertainty. A supposedly easy task "intersects" with the large uncertainty prior distribution at higher probability than when uncertainty is small. When the expectation is high, it is more rational to expect that an assistant exhibiting large uncertainty may fail to cope with easy tasks.

is on the left side of the dashed line. In this situation, large uncertainty (the flatter distribution) portends higher probability than small uncertainty that the assistant can satisfy the intention. Conversely, when the expectation is high, large uncertainty portends higher probability that the assistant is unable to satisfy the intention.

To sum up, we hypothesized a positive effect of expectation on motivation, and an interaction effect of expectation and uncertainty on motivation. To verify these hypotheses, we designed a two-by-two experiment, where expectation was manipulated within group while uncertainty was manipulated between groups. In Section 3, we explain how we created small versus large uncertainty and high versus low expectation in the experimental settings. We also explain why expectation is better manipulated within group than between groups (end of Section 3.2).

## 2 Method

### 2.1 Experimental design

To verify the hypotheses, we conducted an experiment using Echo Dot and Echo Plus, the smart speakers by Amazon (Fig 3). In the experiment, the virtual assistants had different wake words (names): "Echo" and "Alexa". Participants were told that they were two inherently different assistants; no additional information, such as information regarding capability, was provided throughout the experiment (see Section 3.4 and S1 Appendix). The experiment involved two task sections, followed by a "free-choice period", during which the participants could interact (could also decide not to interact) with the virtual assistants. While the participants believed they were asked to interact with the virtual assistants during the task sections, it was actually the experimenter covertly simulating autonomous responses using the Wizard of Oz method [16]. However, in the ensuing "free-choice period", the Wizard of Oz method manipulation was not used, and the participant could interact with the real Amazon Echo assistants (their different wake words, "Alexa" and "Echo", were retained; the interaction was English-only).

The purpose of the task sections is to shape the participant's prior expectation and uncertainty regarding the two assistants, which is the basis of their action selections once they move on to the free-choice period. The purpose of the free-choice period is to covertly observe any spontaneous interaction initiated by the participant, which serves as the observed behavior measurement of intrinsic motivation [17].

Since we have two variables, uncertainty and expectation, we used a two-by-two design. Participants were randomly assigned to either the small-uncertainty group or to the large-

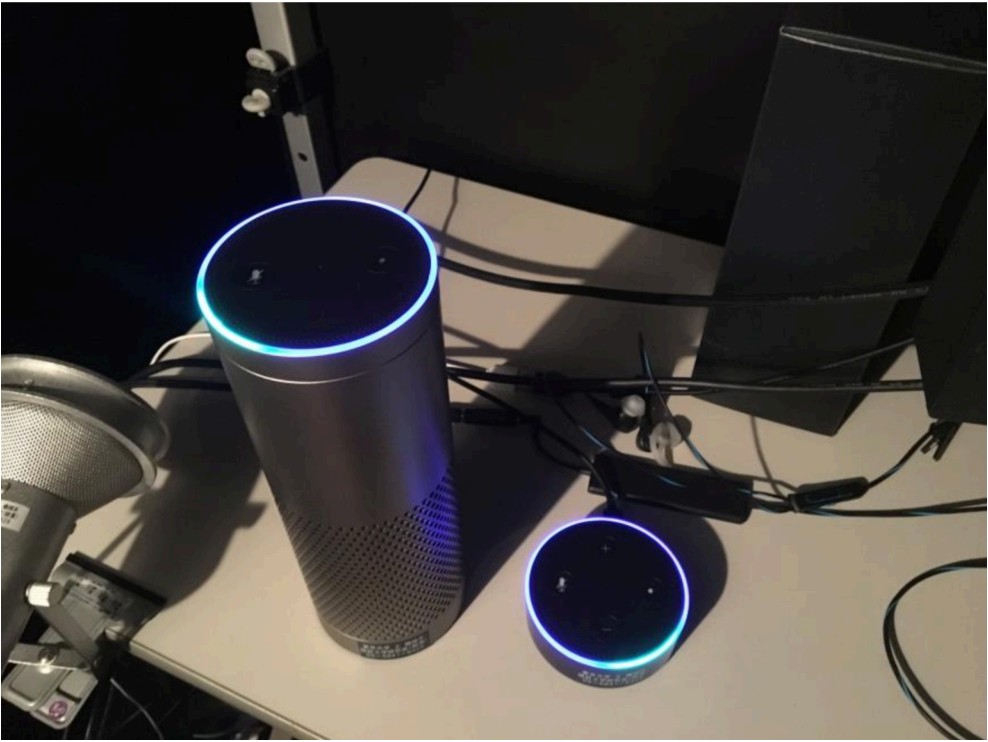

**Fig 3. Smart speakers used for experiment.** Amazon Echo Plus (left) and Amazon Echo Dot (right).

uncertainty group. For each participant, there were two task sections. During each section, the participant was asked to interact with a different assistant and perform interactive tasks designated in a different task list.

The two task lists differed in the overall task difficulty. As we elaborate later in Section 3.2, we designed the tasks and their responses such that, after the two task sections, the participant should conclude that one assistant is more capable than the other. In other words, by the time the participant gets a chance to interact freely (in the free-choice period), we have constructed a high-expectation assistant and a low-expectation assistant. Lastly, when comparing the two groups, the participants in the large-uncertainty group should place less confidence in their expectations (low or high), than those in the small-uncertainty group. In the following paragraphs, we explain how uncertainty and expectation were manipulated.

## 2.2 Manipulation of variables

The Wizard of Oz method allowed us to manipulate the response correctness by the virtual assistants, regardless of Amazon Echo's real capability. Responses during the task sections were audio files generated using the Alexa Developer Console prior to the data collection phase. During the task sections, we streamed the audio files via Bluetooth from the smart speaker.

Task lists and response manipulation details are shown in S1 and S2 Tables. The tasks involved "coin flip", "dice-roll", "get a random number", "judge a number as odd or even", "smart home light bulb maneuver", "converting units", "taking notes", and combinations of them. These tasks were selected because: 1) they were commonly seen functions of real virtual assistants; 2) the responses are time-invariant (unlike weather inquiries) and; 3) the responses are relatively short.

In order to introduce uncertainty, we made the virtual assistants in the large-uncertainty group respond in an inconsistent manner. For instance, we made the assistant cope successfully with the task "Get a random number between 1 and 6". Then, for the next task, "Get two random numbers between 1 and 6", we made the assistant fail (see tasks 4 and 5 in S1A Table and their response manipulation in S2A Table). We assume that these two tasks are of equal difficulty. Therefore, we considered it "inconsistent performance".

To differentiate high- versus low-capability expectation, we made the high-expectation assistant capable of accomplishing a list of difficult tasks, while the low-expectation assistant was made capable of easy tasks but began to struggle when the task difficulty escalated. Difficulty was added gradually: the easy task list (for the low-expectation assistant) primarily consisted of single tasks, whereas the difficult task consisted of successive tasks (do A and then do B), storing variables, calculations using variables, etc. As a recent survey-based study on voice assistant usage revealed, today's voice assistants still cannot "remember context" or "answer several questions at once" [7].

The capability limit of the low-expectation assistant was set at "successive tasks." For instance, the low-expectation assistant was able to "Flip a coin" and "Roll a dice" separately but could not cope with "Flip a coin and then roll a dice". In contrast, the high-expectation assistant was able to cope with "Roll a dice and flip that many coins", which involves calculations using variables. In the experiment, the wake words of the low- and high-expectation assistant were "Echo" and "Alexa", respectively.

In sum, in this experiment, uncertainty was manipulated between groups, while expectation was manipulated within group. Participants from the large-uncertainty group experienced "large uncertainty–low expectation" condition and "large uncertainty–high expectation" condition. Participants from the small-uncertainty group experienced "small uncertainty–low expectation" condition and "small uncertainty–high expectation" condition.

The expectation was manipulated within group because of unwanted learning effects. If we used between-group design for expectation, the two task lists would need to have equally difficult tasks for each group. Meanwhile, the two assistants would still need to show different responses to those virtually identical tasks in order to manipulate uncertainty. If that were the case, participants would expect the same interaction outcomes before the second task section (because they can read task lists before interacting, see Section 3.4 and S1 Appendix), only to be surprised by a different capability profile from the second assistant. Eventually, this would cause participants to have different capability expectations of the two assistants after experiencing both task sections; therefore, we believed expectation must be manipulated within group.

## 2.3 Measurement

Intrinsic motivation was assessed by self-reported and observed behavior measurements. For self-reported measurement, we used questionnaires taken partially from the Intrinsic Motivation Inventory [18] to appraise five aspects of user attitudes toward the virtual assistant: intrinsic motivation of the user, smartness of the assistant, comprehensibility (the extent to which the virtual assistant's thought process is understandable), trust, and human-likeness (see S3 Table). Additional questions were borrowed or modified from previous studies on motivation and technology acceptance [19,20]. For observed behavior measurement, we applied the free-choice paradigm, in which participants are left alone in the experiment room and made to believe that they are no longer under observation. During this free-choice period, participants can initiate interactions of their own decision or do nothing at all [17,21]. The number of

interactions within the free-choice period (which lasts for five minutes) served as an indicator of intrinsic motivation at the behavioral level.

## 2.4 Implementation

The experiment was conducted indoor. The smart speakers Echo Dot and Echo Plus were placed on a table, next to a Philips Hue LED bulb that is mounted on a bulb socket (Fig 3, in the lower left corner). The experimental setting from above is shown in Fig 4.

The overview of experimental procedures can be found in Fig 5. Instructions were given on how to interact with the virtual assistant. These instructions also presented a coherent cover story, implying that the experiment was part of a user-experience research project. The experiment consisted of two task sections, the second one was followed by a 5-muinute free-choice paradigm period. To start the free-choice period, the experimenter used the excuse of needing to print extra documents in order to justify leaving the room. Before leaving, the experimenter covertly reactivated the microphones of the smart speakers, which meant that the participants could interact with the real Echo Dot and Echo Plus during the free-choice period. Interactions during the free-choice period were recorded by the experimenter's computer microphone. The detailed experimental procedures are documented in S1 Appendix.

Thirteen students from the University of Tokyo participated in the experiment (4 female, 9 male, age range from 21 to 26, experiment was conducted in November and December 2018). One set of data had to be discarded because devices lost connection during the experiment. All participants were novices at virtual assistant usage. (We screened potential participants in this way to avoid the difficulty of quantifying different levels of prior user experience, which would introduce noise in the measurement of expectation and uncertainty.) Participants were

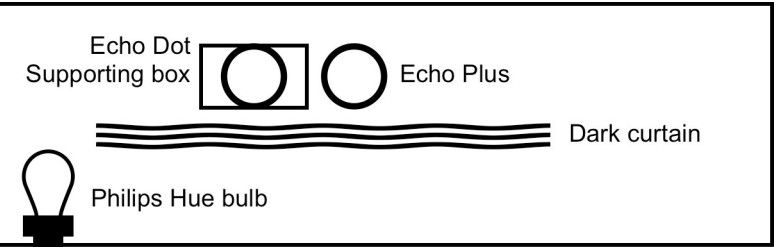

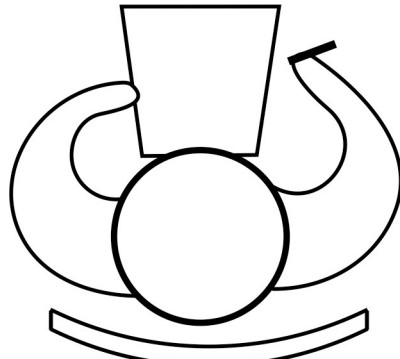

**Fig 4. Overhead view of experimental setting.** The participant sits 50 centimeters (19.69 inches) away from the table. Echo Dot is placed on a supporting box so that both speakers are at the same altitude.

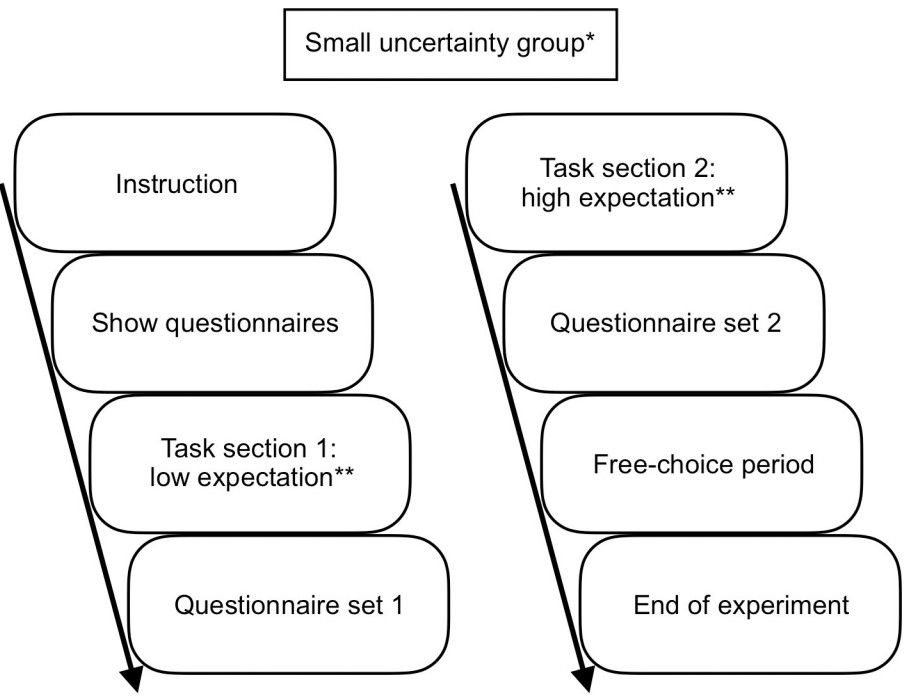

**Fig 5. Overview of experimental procedures.** Uncertainty was controlled between groups, whereas expectation was controlled within each subject.

rewarded with 1000 Japanese Yen (8.8 US Dollar) for participating. They were informed of the monetary reward when recruited and received the reward after the experiment. The study protocol was approved by the Ethics Committee of the Graduate School of Engineering, the University of Tokyo. All participants provided written informed consent prior to their participation in this study.

## 3 Results

We used two-way ANOVA to analyze the effects of expectation and uncertainty on self-reported intrinsic motivation and other sub-scales (see S3 Table and S1 Spreadsheet). The goal was to examine whether there is an interaction effect of the two variables. We used IBM SPSS Statistics 25 (SSTYPE(3), Bonferroni pairwise comparison). Results are shown as bar graphs in Figs 6–10.

Fig 6 shows the self-reported intrinsic motivation scores. We observed no interaction effect of expectation and uncertainty on self-reported intrinsic motivation [$F_{(1,10)}$ = 1.827, p = 0.206]. Instead, we observed a marginally significant main effect of expectation [$F_{(1,10)}$ = 4.095, p = 0.071]; self-reported intrinsic motivation scores were higher under high-expectation conditions. No main effect of uncertainty was observed [$F_{(1,10)}$ = 0.211, p = 0.656]. Fig 7

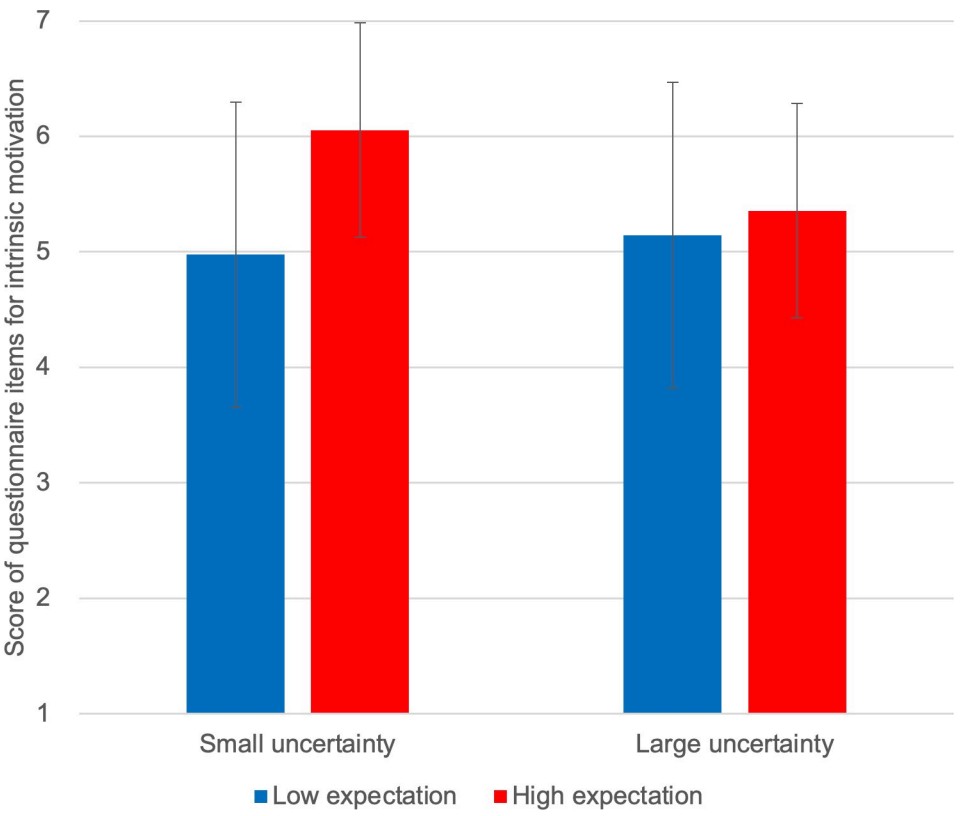

**Fig 6. Self-reported intrinsic motivation.**

shows the self-reported smartness. No interaction effect was observed [F(1,10) = 0.259, p = 0.573]. A significant main effect of expectation was observed on self-reported smartness [F (1,10) = 5.635, p = 0.039]. There was no main effect of uncertainty [F(1,10) = 0.117, p = 0.740]. Fig 8 shows the self-reported comprehensibility scores. Interaction effect was not observed [F (1,10) = 0.002, p = 0.963]. No main effect of expectation [F(1,10) = 1.905, p = 0.198] or of uncertainty [F(1,10) = 0.000, p = 0.983] has been found. Fig 9 shows the self-reported trust scores. No interaction effect [F(1,10) = 1.763, p = 0.214] and no main effects of expectation [F (1,10) = 1.493, p = 0.250] or uncertainty [F(1,10) = 1.179, p = 0.303] were observed. Fig 10 shows the self-reported human-likeness scores. Interaction effect was not found [F(1,10) = 2.361, p = 0.155]. No main effect of expectation [F(1,10) = 0.874, p = 0.372] or of uncertainty [F(1,10) = 0.878, p = 0.371] was found.

Audio files recorded during the free-choice period served as observed behavior measurements. We counted the numbers of interactions observed during free-choice periods. Only complete exchanges, which consist of an initiation by the user and a response by the assistant, are counted. Eight out of twelve participants engaged in a spontaneous interaction; five of them were from the small-uncertainty group. The total number of interactions by the members of the small-uncertainty group was 33, and by members of the large-uncertainty group was 22. Participants initiated a total of 14 interactions with the low-expectation virtual assistant and 41 interactions with the high-expectation virtual assistant. Fig 11 presents the numbers of interactions sorted by uncertainty and expectation.

The observed interactions can be classified according to the type of motivation. We classified interactions as either intrinsically or non-intrinsically motivated. Recent survey-based

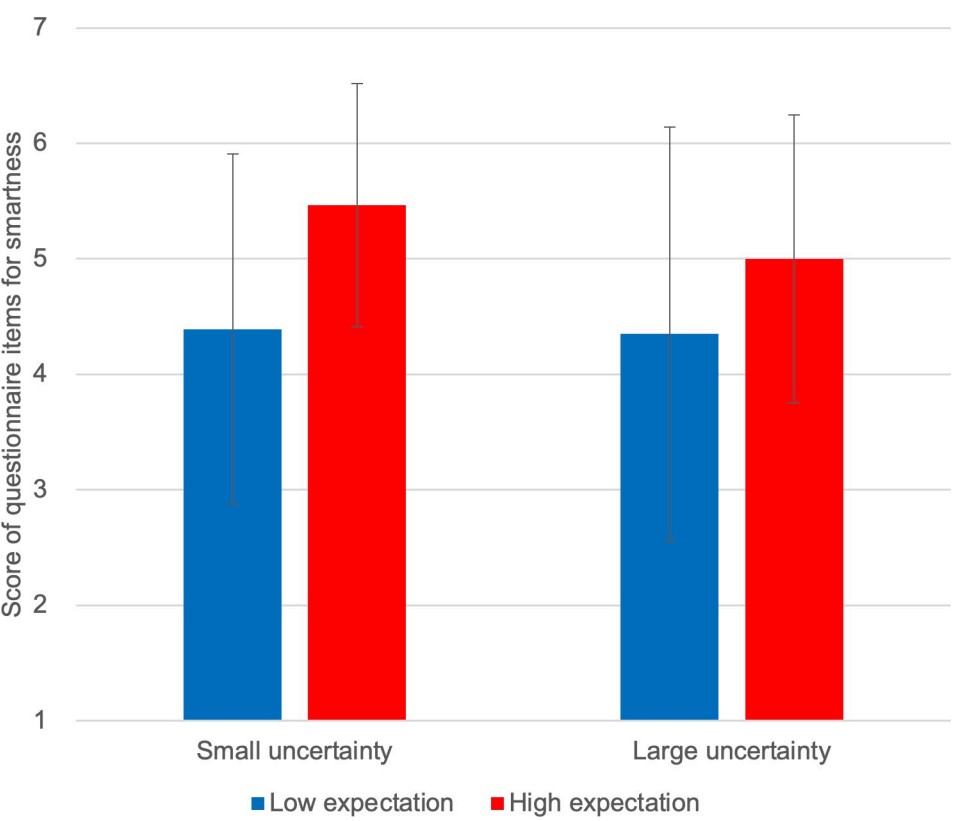

**Fig 7. Self-reported smartness.**

studies on motivation of using voice assistant found that users use voice assistants for fun or to avoid boredom; such activities include "asking for a joke, taking a quiz, reading a poem out loud, and playing social games with friends". [1,2,7]. These activities were classified as "motivated by hedonic benefits" in the literature (in contrast with "'utilitarian benefits" [1]). Although different terminologies were used, it is clear that the above-mentioned activities are driven by the need for enjoyment or pleasure, and therefore should be classified as intrinsically motivated. Intrinsically motivated interactions we observed during free-choice period included pastime and playing activities, which includes game requests, music requests, chatting (e.g., asking the assistant's favorite color), and trivia questions.

Non-intrinsically motivated interactions encompass epistemically and extrinsically motivated interactions. Extrinsically motivated interactions are goal-oriented or efficiency-oriented, pragmatic, utilitarian uses of the assistants [1,7]. These usages are useful and convenient for the user [2]. Epistemically motivated interactions were not explicitly discussed in the above-mentioned literature. Friston et al. [12] related extrinsic and epistemic motivation to "exploitation and exploration", respectively. Exploitation means executing a pragmatic action which fulfills goals directly ("I want the virtual assistant to do this for me"); exploration discloses information that enables pragmatic action in the long run ("What can I do with the assistant?").

In our experiment, we observed epistemically motivated interactions such as test and trial activities, which included attempting unaccomplished tasks from the task sections and exploring the assistants' capability using consecutive challenges (e.g., "My name is. . ." followed by "Do you know who I am?"). We argue that these activities are driven by the need to understand the virtual assistants' limitations, namely, capability.

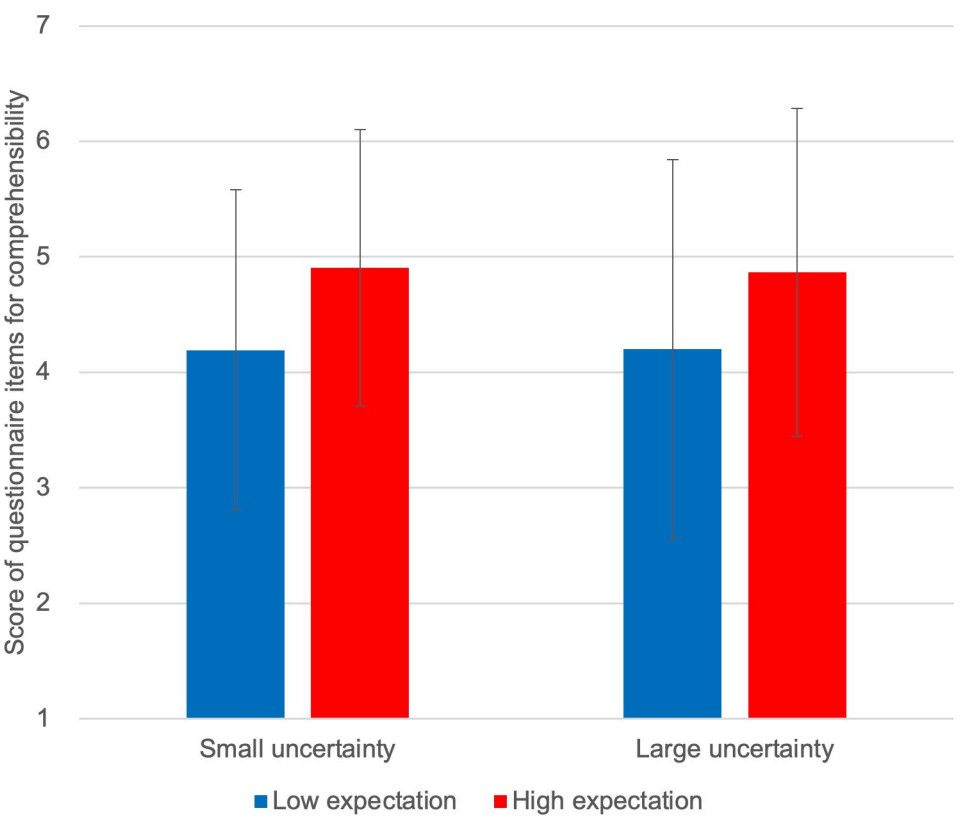

**Fig 8. Self-reported comprehensibility.**

We also observed extrinsically motivated interactions, for instance, searching restaurants ("Find a sushi place near me"). We argue that participants initiated such interactions to fulfill pragmatic goals.

So far, we have used specific, interaction content-based criteria to classify our observations. For the sake of experimental reproducibility, in case replication experiments observe activities beyond our classification, we propose the generalized criteria as follows. Criteria for intrinsically motivated interactions are: 1) the outcome of interaction does not convey new knowledge or information regarding the assistant's capability or limitations; and 2) the interaction does not increase productivity or efficiency.

In light of the above-discussed classification criteria, participants in the small-uncertainty group engaged in 25 intrinsically motivated interactions but only 8 non-intrinsically motivated interactions. The corresponding numbers for participants in the large-uncertainty group were 5 and 17 (see Figs 12 and 13).

We have made box plots of the numbers of interactions by motivation type and uncertainty group under both expectation conditions (Figs 14 and 15). In Fig 14, a reversed proportion of motivation types can be observed. The median numbers of intrinsically motivated interactions for the large- and small-uncertainty groups were two and four, respectively. On the other hand, the median number of non-intrinsically motivated interactions for the large-uncertainty group was two, while for the small-uncertainty group, it was one.

We used a paired sample t-test to compare numbers of interactions by motivation type (parametric data, single variable). For the small-uncertainty group, $t(4) = 2.132$, $p = 0.002$, indicating more engagement caused by intrinsic motivation than by non-intrinsic motivation. For

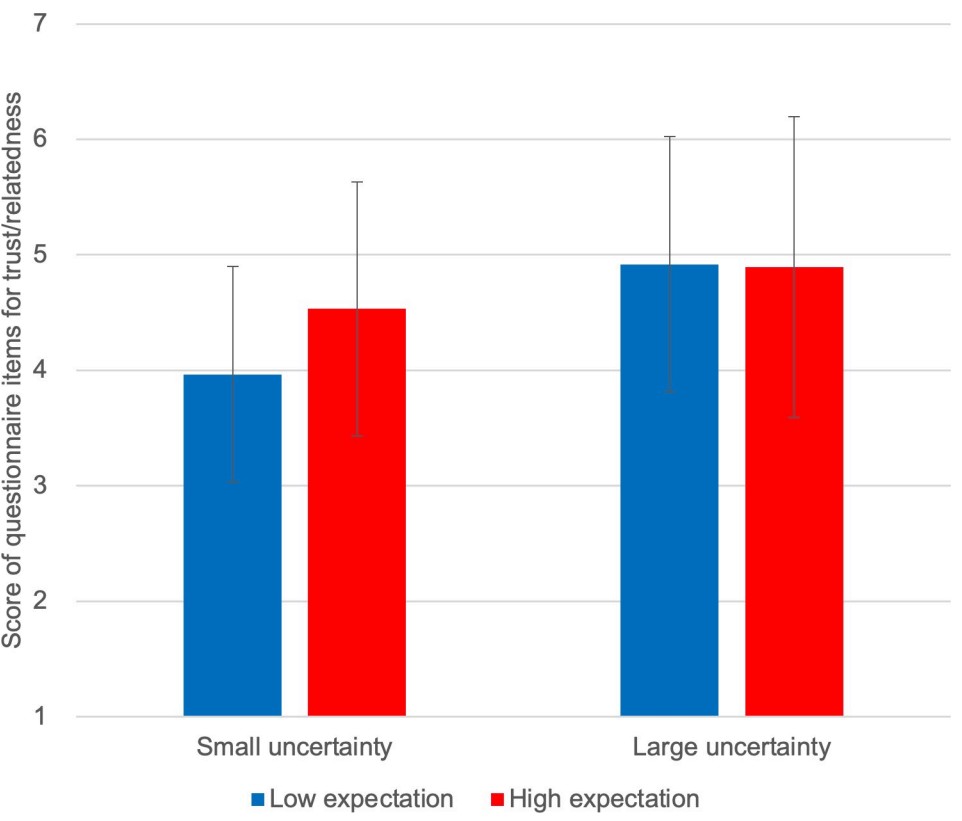

**Fig 9. Self-reported trust/relatedness.**

the large-uncertainty group, only three participants engaged in any interaction at all, which caused insufficient significance: t(2) = 2.919, p = 0.211. Nevertheless, those three participants initiated more interactions for non-intrinsic than for intrinsic motivation. We were unable to conduct the same comparison under low-expectation conditions because only three participants engaged in an interaction, of whom only one was from the large-uncertainty group.

## 4 Discussion

The self-reported results, taken together, did not support our hypothesis of an interaction effect. However, our findings in the observed behavior measurement are promising (see Fig 11). During the free-choice period, participants from both uncertainty groups preferred to interact with the virtual assistant with high capability expectation (41 interactions) than with the low-expectation one (14 interactions). Moreover, participants in the small-uncertainty group had more interactions (33) than participants in the large-uncertainty group (22).

In interpreting Figs 14 and 15, it must be borne in mind that all the participants were given the same amount of time during the free-choice period. Two types of motivation must have been competing against each other. We inferred that, after experiencing large uncertainty, participants tended to act out of non-intrinsic motivation. Conversely, if the uncertainty was small, participants tended to act out of intrinsic motivation.

The reversed motivation type seen in Fig 14 was the result of different action selection strategies. According to the active inference theory [12], when a precise goal exists, extrinsic and epistemic values dominate action selection situationally. Friston et.al. used the term "intrinsic" as a synonym of "epistemic." While extrinsic value is an action's pragmatic potential to reach the

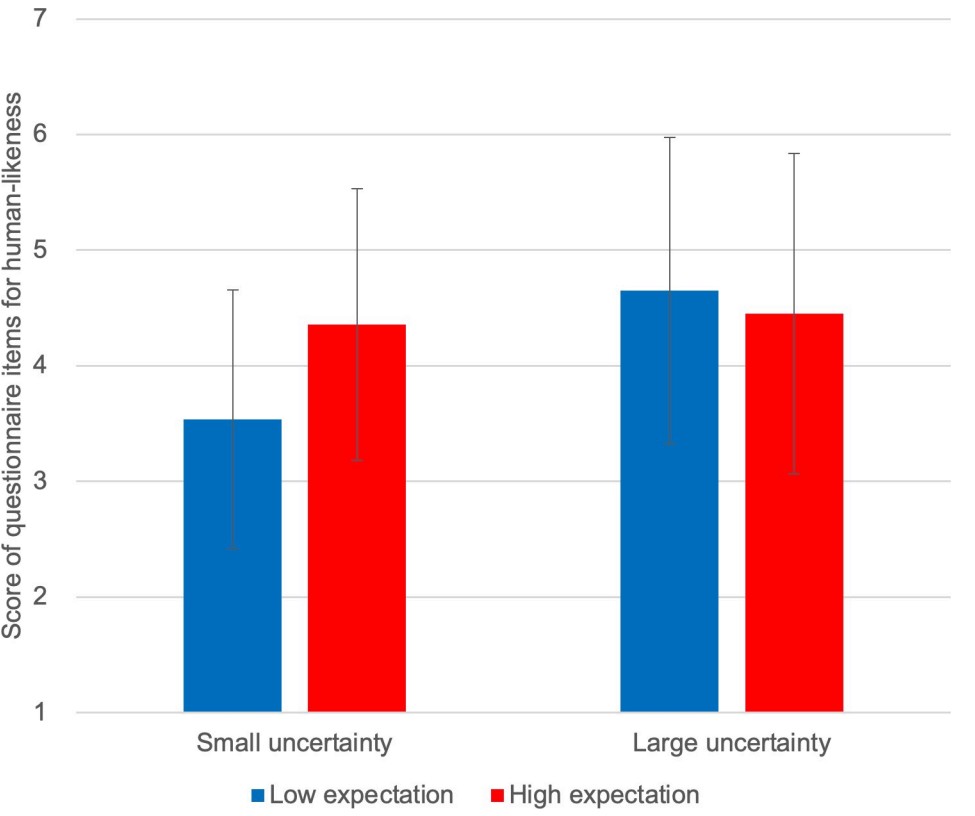

**Fig 10. Self-reported human-likeness.**

goal, epistemic value is defined as an action's potential to eventually enable pragmatic actions. According to the theory, uncertainty "about the state of the world" affects action selection strategy. When uncertainty exists, epistemic value dominates action selection; in other words, humans always prioritize reducing uncertainty through exploration. Conversely, when there is no uncertainty, one tends to act out of extrinsic motivation to fulfill precise goals. In the latter situation, action policies no longer differ in epistemic value, since uncertainty cannot be further reduced; therefore, extrinsic value dominates action selection. On the other hand, when there is no precise goal, Friston et al. [12] argued that epistemic value can dominate action selection. Fig 16 shows the model of action selection derived from active inference theory [12].

Though successful in explaining epistemically versus extrinsically motivated actions, this standard model left unanswered the question of which type of value would dominate action selection in the absence of precise goals. To answer this question, we revised the model based on our observations in the free-choice period. Our revised model is shown in Fig 17. Even in the absence of precise goals, participants can still perceive uncertainty regarding the virtual assistant's capability. Beliefs about the assistant's capability and the uncertainty surrounding it had been established by the learning process (task sections) and would linger on to affect future action selection strategies. Participants who desired further interaction but had perceived a large uncertainty needed first to reduce this uncertainty and naturally undertook test and trial activities.

## 5 Verification experiment of effects of uncertainty

In order to verify the revised model, we created two levels of uncertainty, while encouraging relatively high expectation under both conditions. Instead of faking inconsistent performance, we proposed a method to reduce uncertainty from the baseline level.

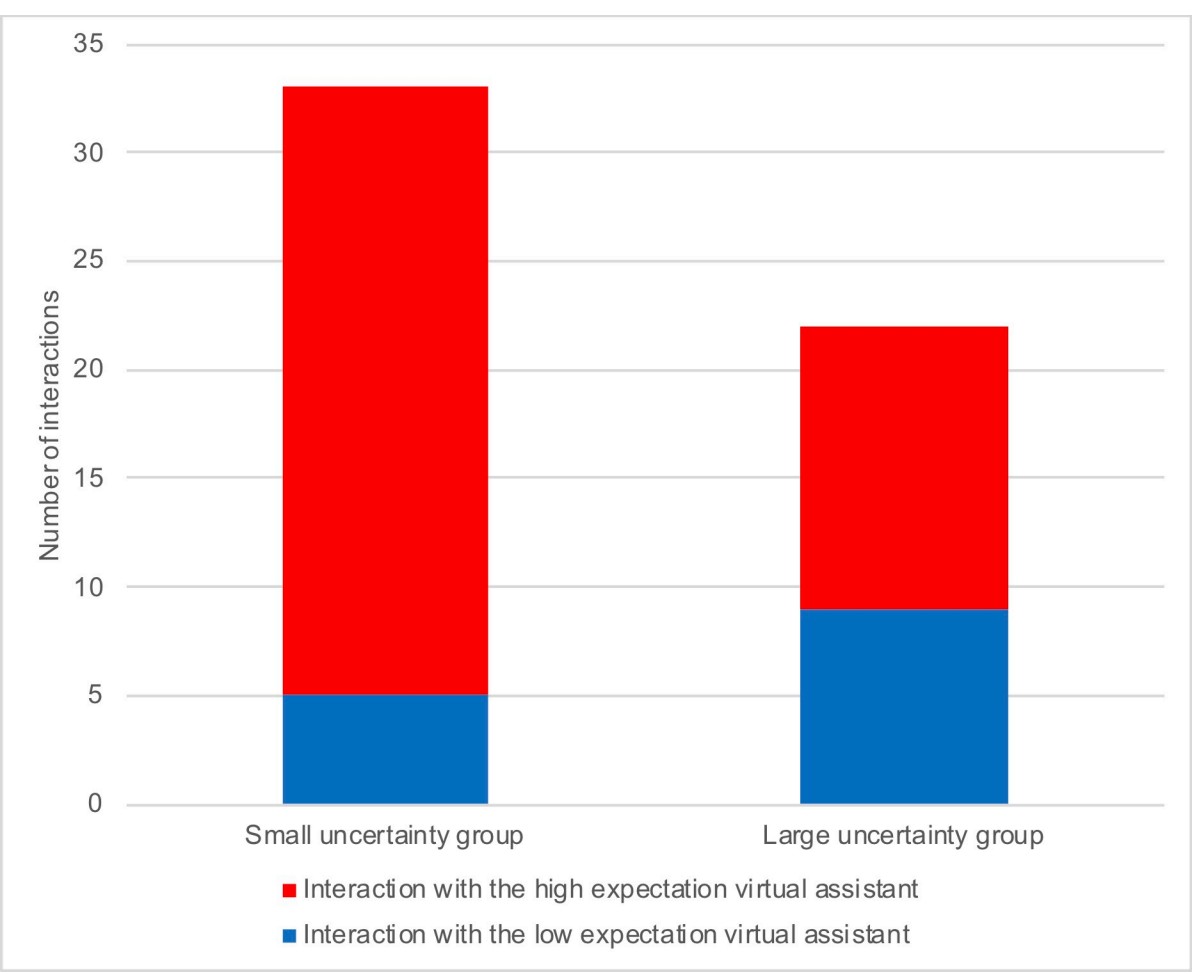

**Fig 11. Accumulated numbers of interactions by uncertainty and virtual assistant in charge.** Note that the participants could choose freely to interact with either assistant or to avoid any interaction.

## 5.1 Manipulation of uncertainty

Consider a scenario where the virtual assistant appears unable to cope and produces an ambiguous response such as: "Sorry, I don't know." Since the response does not disclose causes of the failed task, it can be difficult for the user to adjust their strategies and utterances in future interactions. In some cases, the virtual assistant is able to recognize the user's utterances, but the task is beyond its capability. An example is the task of "doing multiple tasks at once". Alexa does not disclose its limitation to the user (cannot do successive tasks); the user has to determine it through trials. We argue that uncertainty about the (expected) capability can be reduced by disclosing causes of failed tasks, that is, by issuing an informative response instead of an ambiguous one.

In this experiment, the only variable is uncertainty, with two levels, small and large; therefore, we manipulated uncertainty within group. All participants were asked to interact with the small-uncertainty assistant in one of the task sections and interact with the large-uncertainty assistant in the other (order counterbalanced). The wake words were "Alexa" and "Echo" for the small- and large-uncertainty assistant, respectively. The experimental procedures were the same as in the first experiment.

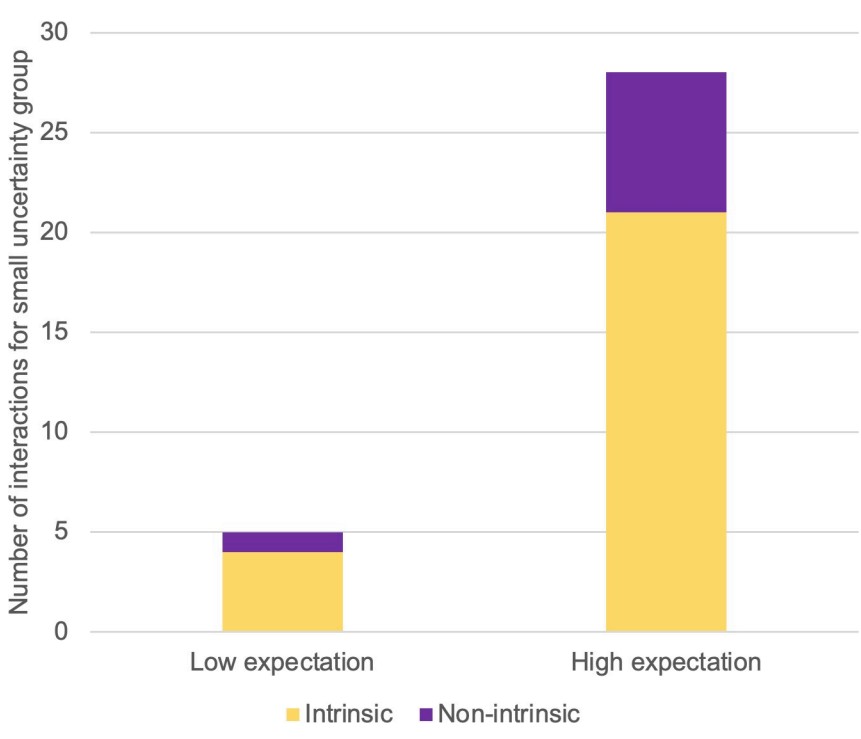

**Fig 12. Numbers of interactions for small-uncertainty group, sorted by motivation type.**

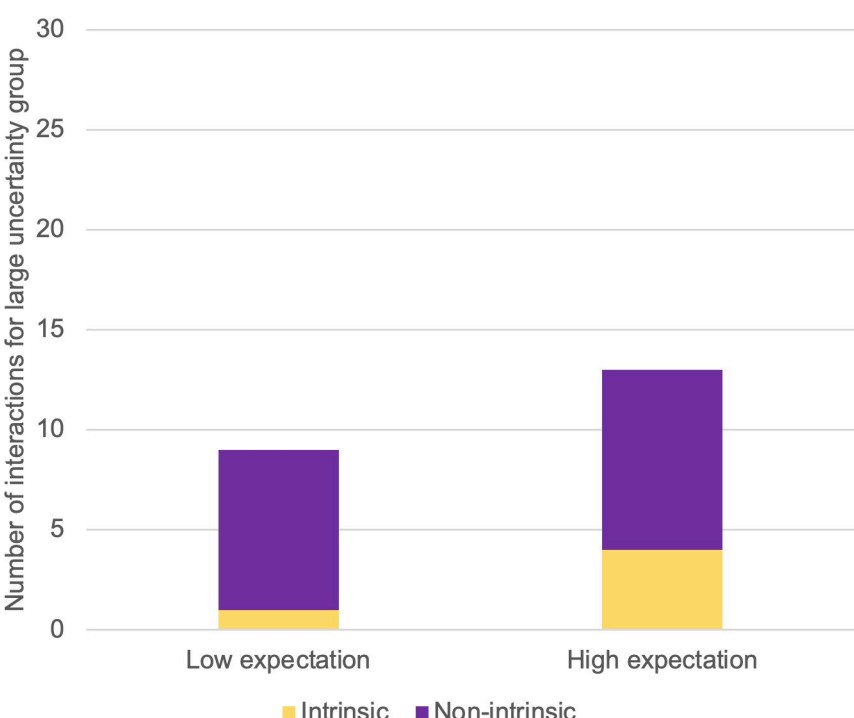

**Fig 13. Numbers of interactions for large-uncertainty group, sorted by motivation type.**

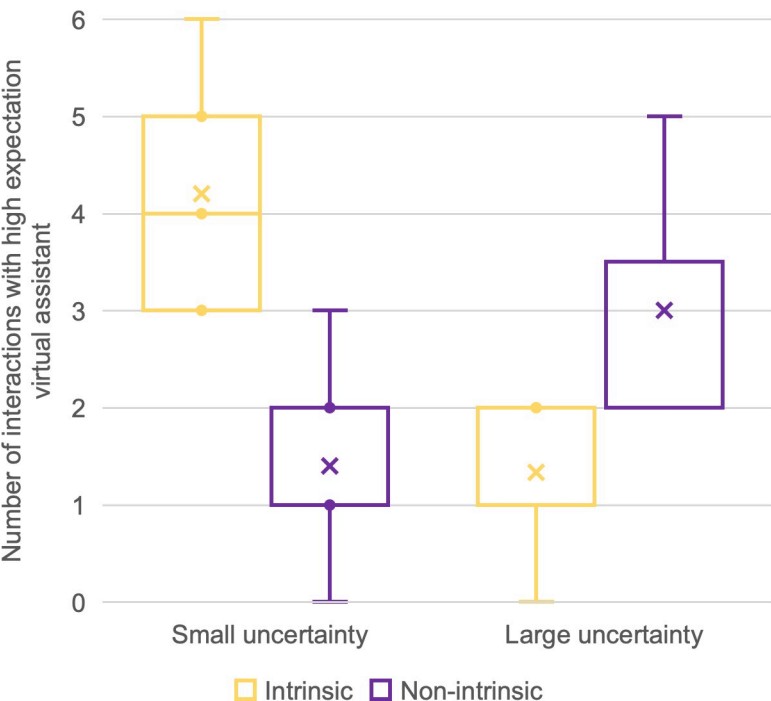

**Fig 14. Number of interactions with the high-expectation virtual assistant.** Sorted by uncertainty group and motivation type.

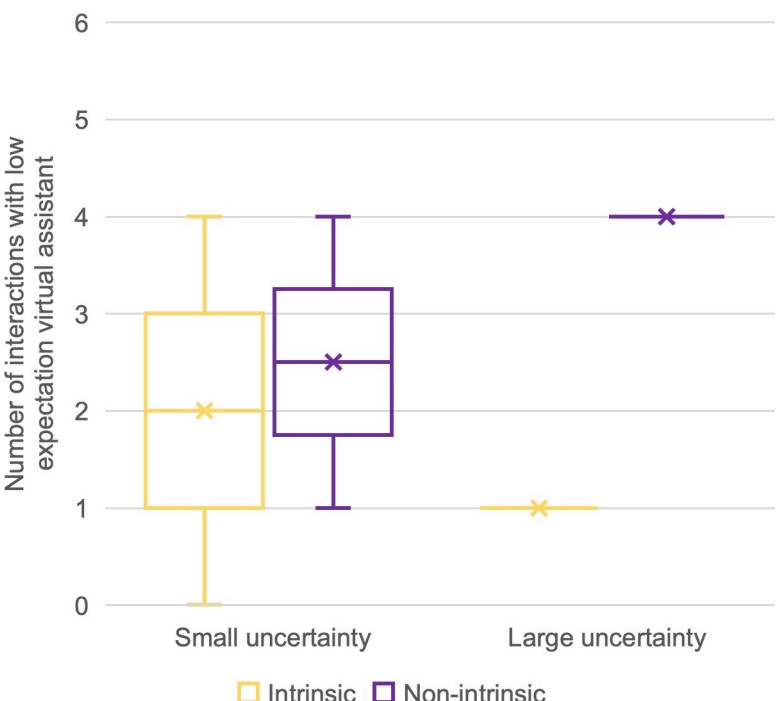

**Fig 15. Number of interactions with the low-expectation virtual assistant.** Sorted by uncertainty group and motivation type.

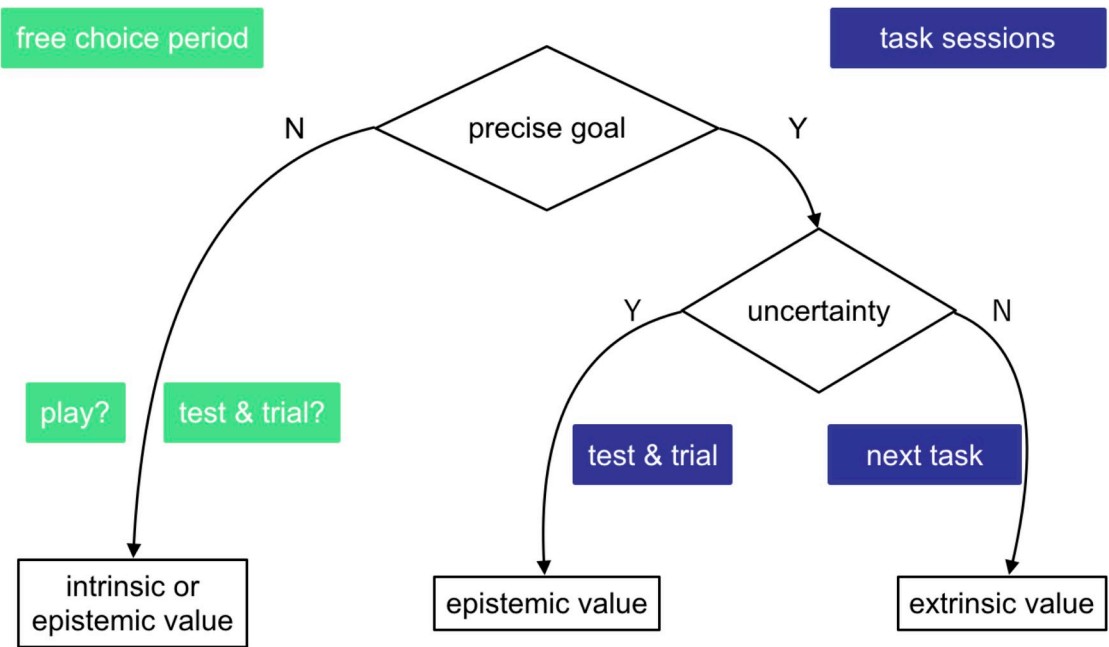

**Fig 16. The original model to explain action selection, from active inference theory [12].** In the presence of a precise goal, action selection is dominated by epistemic value (when uncertainty is great) or extrinsic value (when uncertainty cannot be further reduced).

The small- and large-uncertainty task lists are documented in S4 Table. In the large-uncertainty task list, the virtual assistant only issues ambiguous responses when it is unable to perform the task. Conversely, in the small-uncertainty task list, informative responses were

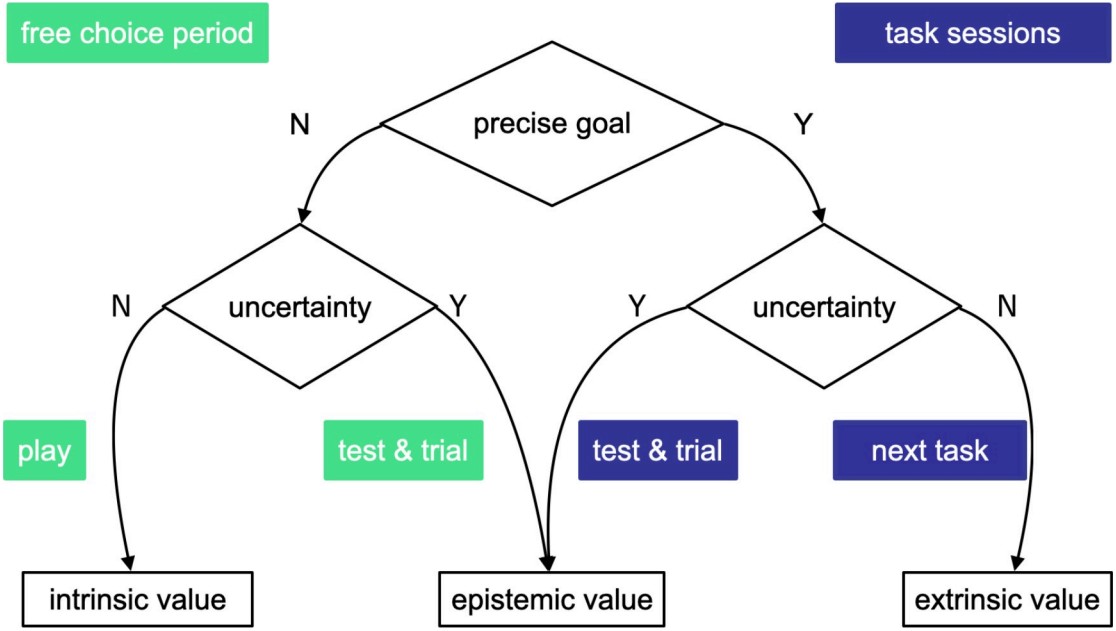

**Fig 17. Revised model of action selection.** Even in the absence of precise goals, uncertainty perceived during previous interactions should affect action selection.

issued. For instance, when asked to "calculate 1000 US dollars minus 70,000 Japanese Yen," we prompted an informative response: "Sorry, I can't calculate between different currencies." Because the responses of the two assistants did not differ in terms of their capability profile, we argue that the expectations induced by the two task sections should be identical.

## 5.2 Implementation

The experimental setup and procedures, including instructions, were identical to those of the first experiment (see Section 3.4 and S1 Appendix). The questionnaires used can be found in S5 Table. To counterbalance the effects of the order, half the participants started in the large-uncertainty condition, and half in the small. The experimental procedures are detailed in S1 Fig.

Ten students from the University of Tokyo (4 female, 6 male, age range from 21 to 27, experiment conducted in July 2019) participated in the experiment. One set of data had to be discarded because devices lost connection to the Internet during task sections. All participants were new to virtual assistants, and none of them had participated in Experiment 1. Participants were rewarded with 1000 Japanese Yen for participating. They were informed of the monetary reward when recruited and received the reward after the experiment. The study protocol was approved by the Ethics Committee of the Graduate School of Engineering, the University of Tokyo. All participants provided written informed consent prior to their participation in this study.

## 6 Results

We examined the content of interactions during the five-minute free-choice period. The complete protocols can be found in S6 Table. Eight of the ten participants engaged in spontaneous interactions during the free-choice period. Only two participants made conversation with the large-uncertainty virtual assistant; the other participants only interacted with the small-uncertainty virtual assistant. As in Experiment 1 (and as we had predicted), the conversation contents could be sorted into categories. Testing and making trials were regarded as evidence of non-intrinsic motivation; playing and pastime activities were regarded as evidence of intrinsic motivation. We excluded conversations that were interrupted or received no response. The number of interactions of both types are presented in Fig 18.

Intrinsically motivated interaction occurred 45 times, while non-intrinsically motivated interaction occurred 11 times. The large-uncertainty virtual assistant did not draw the attention of six of the nine interacting participants at all.

We used a paired sample t-test to compare the numbers of interactions targeting the large- and small-uncertainty virtual assistants. The results showed that participants interacted with the small-uncertainty virtual assistant more than with the large-uncertainty one. The difference had statistical significance: $t(8) = 3.1724$, $p = 0.007$.

Next, we used a one-tail paired t-test to compare the numbers of intrinsically versus non-intrinsically motivated interactions targeting the small-uncertainty virtual assistant. The results showed that participants had more intrinsically motivated interactions than non-intrinsically motivated ones. The difference had statistical significance: $t(8) = 3.5228$, $p = 0.004$. The results are shown in the box plot in Fig 19. For completeness, we remark that the mean of the number of interactions with the small-uncertainty virtual assistant was 5.67, with a standard deviation of 4.8218.

On the other hand, when the interaction partner was the large-uncertainty virtual assistant, we could not conclude which type of motivation caused more interaction: the p-value was 0.2971. However, the total number of interactions was considerably smaller (mean = 0.56, standard deviation = 1.0138).

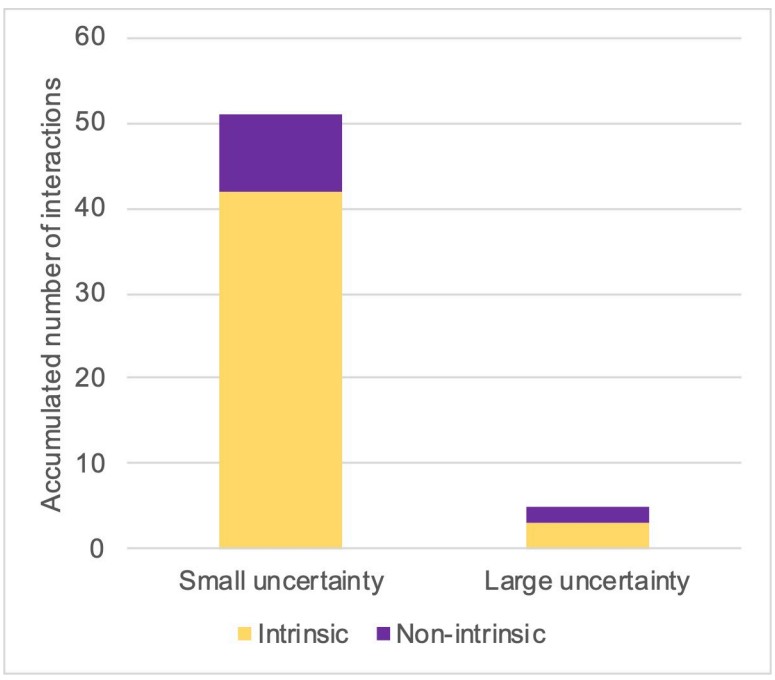

**Fig 18. Accumulated number of interactions by motivation type.**

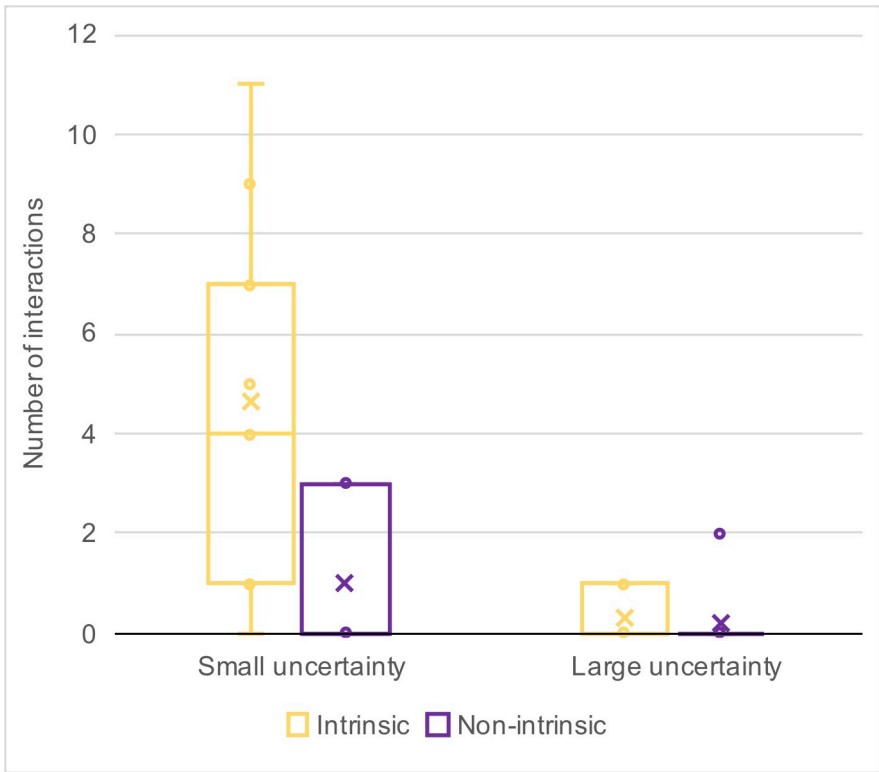

**Fig 19. Average number of interactions by uncertainty and motivation type.**

We used a paired sample t-test to compare self-reported intrinsic motivation, smartness, and comprehensibility under different uncertainty conditions (see Fig 20, also see Spreadsheet S5 for minimal dataset). No significant difference was found between the self-reported intrinsic motivation scores with small and large uncertainty [$t(8) = 1.309$, $p = 0.113$]. However, small uncertainty resulted in significantly higher smartness [$t(8) = 4.304$, $p = 0.0013$] and comprehensibility [$t(8) = 2.9493$, $p = 0.009$].

## 7 Discussion

In this case, our hypothesized model (Fig 17) was validated by experimental findings on the behavioral level. We successfully fostered intrinsically motivated interaction by issuing informative responses that explained the reasons for failure. As shown in Fig 18, the virtual assistant with small uncertainty won the participants' attention. Participants on average interacted 5.67 times with the small-uncertainty virtual assistant, and only 0.56 times with the large-uncertainty one. The difference is significant, with $p = 0.007$. Furthermore, as can be seen in Fig 19, participants initiated more interactions for intrinsic intentions than for non-intrinsic intentions when targeting the small-uncertainty virtual assistant ($p = 0.004$).

We can conclude that, when operating under high-expectation conditions, reducing uncertainty is an effective way to foster intrinsically motivated interaction. When the user expects the virtual assistant to cope, but it fails, uncertainty immediately rises because of the unexpectedness of such behavior. However, an informative response soon afterwards serves to decrease this uncertainty, since it removes the user's uneasiness about the hidden causes of the unexpected event. For a short period, and as long as the user's attention is still directed on the

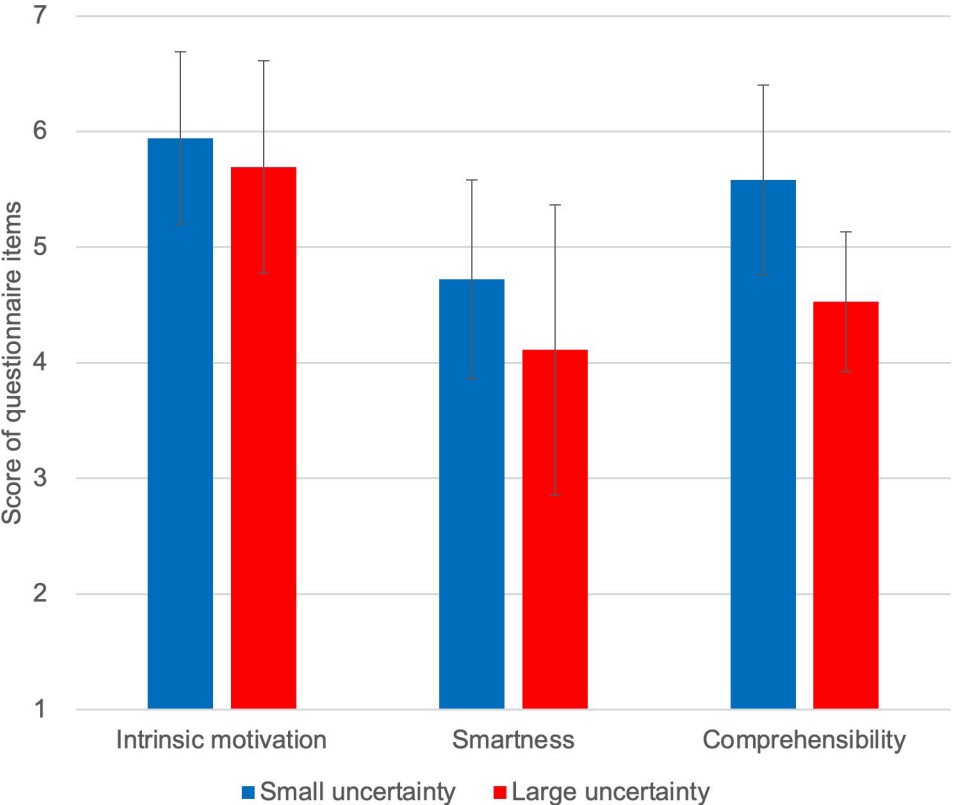

**Fig 20. Self-reported intrinsic motivation, smartness, and comprehensibility.** Sorted by uncertainty group.

interaction, there is no more uncertainty to reduce. As we expected, such absence of uncertainty is important for determining whether intrinsic or epistemic value dominates in action selection, and thus whether interactions are spontaneous.

# 8 General discussion

## 8.1 Mechanism of intrinsically motivated interaction

We constructed a theory to discuss user motivation and spontaneous interactions in the context of conversing with virtual assistants. Our theory withstood experimental examination: we proved that when the user has high expectations of capability, intrinsically motivated interaction can be fostered by reducing uncertainty (more specifically, by making the virtual assistant issue an informative response whenever it is unable to satisfy the user's intention).

The mechanism behind our findings is best explained by focusing attention on three interlocked aspects: motivation, expectation, and action selection. Motivation depends on the expectation of capability: it will result in spontaneous interaction only when the user anticipates that the virtual assistant will successfully cope with the stated intention.

We emphasize that user motivation should be divided into two categories: intrinsic and non-intrinsic. Intrinsic motivation is derived from the expected enjoyment or pleasure of the action itself. Non-intrinsic motivation is derived from the need to achieve an external goal and should be further divided into epistemic and extrinsic motivation.

In general, if the user is exploring whether the assistant is capable of doing something, the action should be considered epistemically motivated. If the user is certain about the capability and the interaction promises pragmatic value in terms of fulfilling separable goals by the user, it should be seen as extrinsically motivated. Borrowing terminologies from the active inference theory, the former case counts as exploration, which enables future pragmatic actions; the latter counts as exploitation, which fulfills the goals directly [12].

In the context of virtual assistant interaction, intrinsically motivated interactions involve activities such as pastime and playing activities. Epistemically motivated interactions involve testing and making trials ("What can I do with the assistant?"). Extrinsically motivated interactions involve pragmatic usage to gain information or have some "chores" done by the assistants instead of the users themselves, thus saving time [1]. For clarity, using the virtual assistant as a learning tool counts as extrinsic usage because the separable goal is to learn.

Spontaneous interactions can be intrinsic as well as non-intrinsic. To ask when actions are intrinsically motivated is to ask how they are selected. Action selection, according to reversal theory [22], depends on the user's state of mind. The human mind can operate in two meta-motivational modes: the goal-oriented telic mode and the activity-oriented paratelic mode. Intrinsically motivated interaction is most likely to occur in the paratelic mode. The mind shifts between these two modes situationally; such shifts are termed "reversals". Our work shows that an informative response can facilitate reversal from the telic to the paratelic mode.

Issuing an informative response is merely one of many methods to foster intrinsically motivated interaction; it works by reducing uncertainty, so that the user is placed in a comfortable condition (paratelic mode), free from pressure. However, one still might ask why reducing uncertainty is the key to intrinsic motivation.

Imagine being trapped in a jungle and facing dire survival concerns. Clearly, actions are not likely to be taken for enjoyment of the activity itself. One must explore the environment, in other words, reduce uncertainty, before deciding which way to go as well as where to find food. Interestingly, the human mind can actually derive pleasure from resolving uncertainty; designers and artists have long used this trick to add novelty to their work. In the theory of aesthetic valence, it is pointed out that optimum novelty requires work to be perceived initially as

a chunk of incomprehensible information (which introduces uncertainty) [23]. Artists and designers weave hints and cues that facilitate sense-making into the novel work, so that it does not take observers too long to arrive at an understanding of the work. The pleasure perceived when the work finally makes sense is termed the "second reward" [23], as opposed to the first reward from simply receiving the information. The informative response strategy works the same way.

An informative response would most likely start with "Sorry, I can't/I don't know. . ." The user is notified by the "Sorry," knowing that their intention is not to be satisfied. Then, the informative part explains "what went wrong" or "where was the limit," helping the user make sense. It does not only offer consolation but also provides the user with hints for future interaction. For instance, after hearing that the assistant cannot perform multiple tasks at once, the user may adjust interaction strategies to simplify a complex intention into single tasks. For example, recent studies on explainable AI stressed the importance of clarifying the assistant's limitations, so that users understand what interaction choices they have [7,24,25].

Studies on the relationship between physiological arousal and hedonic states have also stated that reducing uncertainty "makes the world enjoyable" [26,27]. Certainly, this does not mean that, from the hedonic point of view, an informative failure response would be better than a coping response. As we observed in experiments, high performance by the virtual assistant leads to high expectation of capability, resulting in more intrinsically motivated interactions. Nevertheless, if we wish to apply such knowledge to foster intrinsically motivated interaction, it is effective to create a second reward when the user's intention cannot be satisfied, by informing of the limitations of the assistant after the inevitable "Sorry".

## 8.2 Application to virtual assistant design

Our experimental findings are transferable to practical virtual assistant design. Modern virtual assistants recognize users' intentions by filling slots. Slots are variables that are embedded in applications to serve the user's requests. They are filled by picking up key information from user utterances. For instance, a flight booking application typically expects to fill slots such as "destination" and "date". For Amazon Echo series, developers are allowed to define intents and slots in order to capture meaningful information from user utterances. For each intent and under each slot there can be multiple corresponding utterances, which are spoken phrases with a high likelihood of being used to convey intent [28].

If the assistant picks up an utterance that is understandable but not implementable, it should issue an informative response. To achieve this, developers should collect utterances that are recognized but with which the assistant cannot yet cope. Next, they need to categorize the reasons for this failure and try to obtain clusters of such reasons. Each cluster should then be addressed by a set of (at least one) informative responses explaining to the user that the virtual assistant cannot cope with the intention yet. Addressing ambiguous responses cluster by cluster is not an exhaustive method, but it does not take an exhaustive method to reduce uncertainty. For instance, telling users "I'm not able to do two things at once" actually forestalls a very large set of problems that would be impossible to list exhaustively.

## 8.3 Limitations

A major limitation of this study is that manipulation of expectation and uncertainty was implemented only in task sections, in which all activities were extrinsically motivated. In addition to our current findings, we wish to investigate results of such manipulation when they are performed during intrinsically motivated or intrinsically motivating activities. In real-life interaction scenarios, the learning process is experienced through tasking as well as through playing.

Moreover, one study found that, in the context of gaming, the uncertainty of success brings suspense, which in turn leads to enjoyment [29]. Indeed, with the current findings we cannot predict how uncertainty during intrinsically motivated interaction will further affect ensuing action selection because the user's acceptance of uncertainty is likely different when in paratelic rather than telic mode.

## Supporting information

**S1 Fig. Procedures for the verification experiment of effects of uncertainty.**
(PDF)

**S1 Table.** A. Task list for low-expectation conditions. B. Task list for high-expectation conditions.
(PDF)

**S2 Table.** A. Response manipulation for low-expectation conditions. B. Response manipulation for high-expectation conditions.
(PDF)

**S3 Table.** A. Questionnaire items. B. Questionnaire items by sub-scales.
(PDF)

**S4 Table.** A. Tasks for small-uncertainty condition (Verification experiment of effects of uncertainty). B. Tasks for large-uncertainty condition (Verification experiment of effects of uncertainty).
(PDF)

**S5 Table.** A. Questionnaire items for the verification experiment of effects of uncertainty. B. Questionnaire items by sub-scales for the verification experiment of effects of uncertainty.
(PDF)

**S6 Table. Protocol of dialog for free-choice period.**
(PDF)

**S1 Appendix. Comprehensive experimental procedures.**
(PDF)

**S1 Spreadsheet. Minimal data set.**
(XLSX)

## Author Contributions

**Conceptualization:** Chang Li, Hideyoshi Yanagisawa.

**Data curation:** Chang Li.

**Formal analysis:** Chang Li.

**Funding acquisition:** Hideyoshi Yanagisawa.

**Investigation:** Chang Li.

**Methodology:** Chang Li.

**Project administration:** Hideyoshi Yanagisawa.

**Resources:** Hideyoshi Yanagisawa.

**Software:** Chang Li.

**Supervision:** Hideyoshi Yanagisawa.

**Validation:** Chang Li.

**Visualization:** Chang Li.

**Writing – original draft:** Chang Li.

**Writing – review & editing:** Chang Li, Hideyoshi Yanagisawa.

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
