## [Decision Letter · Decision Letter 0]

17 Aug 2020

PONE-D-20-16053

Intrinsic motivation in virtual assistant interaction for fostering spontaneous interactions

PLOS ONE

Dear Dr. Li,

Thank you for submitting your manuscript to PLOS ONE. After careful consideration, we feel that it has merit but does not fully meet PLOS ONE’s publication criteria as it currently stands. Therefore, we invite you to submit a revised version of the manuscript that addresses the points raised during the review process.

Two Reviewers have evaluated the manuscript and have given generally favorable opinions. However, revisions were suggested, especially regarding some aspects in manuscript structure, and clarifications to methodology and background/discussion. 

We look forward to receiving your revised manuscript.

Kind regards,

Stefano Triberti, Ph.D.

Academic Editor

PLOS ONE

Journal Requirements:

2.We note that you have indicated that data from this study are available upon request. PLOS only allows data to be available upon request if there are legal or ethical restrictions on sharing data publicly. For more information on unacceptable data access restrictions, please see http://journals.plos.org/plosone/s/data-availability#loc-unacceptable-data-access-restrictions.

3. Please ensure that you refer to Figure 5, 6, 7 and 8 in your text as, if accepted, production will need this reference to link the reader to the figure.

Reviewers' comments:

Reviewer's Responses to Questions

**Comments to the Author**

1. Is the manuscript technically sound, and do the data support the conclusions?

Reviewer #1: Yes

Reviewer #2: Yes

2. Has the statistical analysis been performed appropriately and rigorously? 

Reviewer #1: Yes

Reviewer #2: I Don't Know

3. Have the authors made all data underlying the findings in their manuscript fully available?

Reviewer #1: Yes

Reviewer #2: Yes

4. Is the manuscript presented in an intelligible fashion and written in standard English?

Reviewer #1: No

Reviewer #2: Yes

5. Review Comments to the Author

Reviewer #1: In this manuscript, the authors discuss the role of intrinsic motivation in human-machine interaction. More specifically, they present a study using a virtual assistant to investigate the effects of (a) user's perception of assistant's capability and (b) uncertainty to intrinsic motivation. The topic of intrinsic motivation in human-machine interaction is really interesting and challenging and the manuscript includes useful related work and background in this area. However, there are some improvements and adjustments that should be made to improve the presentation of the paper.

- The authors present the design and results of two experiments; the findings from the first experiment were used to inform the design of the second one, to my understanding. If that is the case, this should be clearly mentioned in the abstract and the introduction sections. Each Experimental section should be then renamed (instead of Experiment 1/2, place a more informative title). Moreover, the group design of these two experiments should be justified. Why each group design was used in each experiment. Is it a 2x2 design, since you investigate the effects of two variables? A justification would also help for comparison and reproducibility.

- The hypothesis is clear and defined based on the theory discussed in Section 2. However, more references are needed to justify e.g., (Line 98) Since the explicit goal of exploration is learning, it can be concluded that uncertainty is responsible for non-intrinsically motivated activities. Moreover, a clear but short description of the tasks used should be also included in the text and not only in the tables. Why such tasks were selected? Maybe parts of Section 3.2.5 could be moved to the introduction as background knowledge.

- Were the participants asked to carry out the tasks in the given order, as presented in Tables S1 and S2? As mentioned in the first comment, the design of the groups must be explicitly defined and clearly described. How were the participants assigned to groups?

- Table 1 is not informative. Speech recognition is not mentioned before that point and its not clear to me what Cases 1-4 are and how they are used in the experiment.

- I am not sure about the categorization of the user utterances into intrinsic and non-intrinsic. for example, "Translate something" could be considered intrinsic or not based on the way it is expressed, or not? e.g., (Can you) translate this for me? Is it test or pastime activity?

- Experiment 2 has to do with explainability. Maybe a few references of Explainable AI and how it affects user motivation should be included.

- The manuscript needs proofreading and many English language edits.

In conclusion, the manuscripts includes useful information and finding in this topic of intrinsic motivation. I believe that if the presentation and some parts of the paper improve, it would strengthen the quality of the manuscript.

Reviewer #2: This study reports on an experiment with 13 students (12 sets of data) in Tokyo in which intrinsic vs. extrinsic motivation was studied by manipulating expectation and uncertainty in relation to tasks carried out with two voice assistants (Amazon Echo and Amazon Dot). Such spontaneous interactions as occurred during five minutes alone with the assistants were also analyzed. Not many significant findings were reported, in part, I suspect, because of the small number of participants and the number of concepts that were manipulated. A follow-up experiment manipulating only uncertainty was then conducted with 10 students (9 sets of data), in which the device provided informative reasons when unable to answer a question; the results of this experiment provided significant support for the study’s hypothesis that intrinsic motivation is related to uncertainty.

This is a complex study, and I had some difficulty following all the threads of reasoning until I reached the second, more simplified, experiment. The theoretical construct as laid out in section 2 involves the interrelations of multiple concepts, and it is described mostly in abstract axioms. It would help to tie the abstractions to possible real-world scenarios, both to clarify and to motivate the construct.

Some terms and concepts are invoked without definition. These include ‘robot,’ ‘complete interaction,’ ‘meaningful interactions,’ and ‘fitting slots.’ Also, the terms ‘subjective’ and ‘objective’ for the data collected in the experiments seem somewhat judgmental. Less value-laden and more descriptive terms would be ‘self-reported’ and ‘observed behavior’.

The Methods seem basically fine, although minor clarifications are needed. The role of the two devices (Echo and Dot) in the experimental setups is not stated as clearly as it could have been, especially for Experiment 1. Also, more information is needed about participant demographics and the times when the experiments were performed.

The Discussion is theoretically informed and clearer than the introductory sections. The comments about the gratifications of uncertainty resolution (“second rewards”) are quite insightful.

The written English is virtually grammatically flawless. However, in the first two major sections of the paper the writing is highly abstract and for that reason, sometimes difficult to follow.

I also have made a number of detailed comments, as follow:

P 2, ll 19-20 “With the growing utility of today’s conversational virtual assistants, the importance of user motivation in human-robot interaction is becoming more obvious.” (I’m used to distinguishing virtual assistants from robots. What is your definition of a robot?)

L 35 (“smartness” = “likeability”? This is not always true for people. Why should we assume it for CVAs?)

P 3, ll 46-48 “Intrinsic motivation should be the next focal issue in virtual assistant interaction (and indeed in HRI generally) because it is crucial for enhancing the likability and the perceived smartness of future smart products.” (The causality goes in the opposite direction, no?)

L 55 “circumstances the user (in general, the agent)” (Calling the user of an AI agent the ‘agent’ is confusing. What’s wrong with ‘user’?)

Ll 56-57 “Eventually, we wish to anticipate and even shape behavioral strategy, thus fostering spontaneous interactions. Such findings would be valuable both for the academic community and for industry.” (Spell out what the value would be.)

L 58 “We define a spontaneous interaction as any action that is taken by agents of their own will.” (In your experiments, ok, but not in general, since users ask Alexa for information for extrinsic motives “of their own will” all the time.)

P 4, ll 76-78 “Normally, after the learning process, agents can re-evaluate the assistant’s capability and their own action strategies by asking whether their commands met with compliance. In the presence of too much uncertainty, such re-evaluation becomes difficult.” (Why is that, exactly? The abstractness of some of the statements in this section makes them difficult to process, e.g., “An interaction suffers from information loss at the interface” and “uncertainty diminishes the effect of expectation.” Perhaps these are truisms in the authors’ discipline, but as a potentially interested reader from outside the discipline I find them puzzling. More explanations and hypothetical examples would help.)

P 5 ll 88-89 “research has shown that an intelligent robot is perceived as more likable [10].” (Independent of anthropomorphism, size, mobility, etc.?)

Ll 91-92 (Your “clarification” here is confusing. Reword or omit.)

Ll 97-99 “Generally, such disruption will lead the agent to re-explore the capability of the virtual assistant. Since the explicit goal of exploration is learning, it can be concluded that uncertainty is responsible for non-intrinsically motivated activities.” (How does this follow logically?)

P 6, Ll 102-107 “Cue-integrating theory states that the larger the uncertainty, the less reliable the evidence [12]. Therefore, the resultant expectation is always closer to the more reliable evidence. When observations suggest that the virtual assistant is of low capability, small uncertainty will draw expectation towards observation, because the observation will be considered accurate. However, large uncertainty reduces the impact of the observations … As a result, we believe that when expectation is low, large uncertainty leads to higher expectation, and thus benefits motivation more than small uncertainty.”

(I’ve read these sentences over several times and still don’t understand them. At least five different concepts are invoked in this subsection (expectation, degree of expectation, uncertainty, degree of uncertainty, motivation [intrinsic and extrinsic]). The relationships among them should be laid out more systematically. “Inconsistent performance by the virtual assistant gives rise to uncertainty;” ok. “Capability expectation is based not only on the agent’s observations, but also on the agent’s prior belief in the assistant’s capability;” ok, although this raises the question: What gives rise to the prior beliefs, if not the agent’s observations? What determines high vs. low expectations in the real world? Also, what determines “small” versus “large” uncertainty?)

Ll 113-116 “we hypothesize that intrinsic motivation is affected by the interacting variables expectation and uncertainty as follows: High expectation has positive effects on intrinsic motivation. Uncertainty has beneficial effects when expectation is low and harmful effects when expectation is high.” (What possible real-world scenarios do these hypotheses map onto?)

P 7, ll 141-142 “S1 and S2 Tables”  “Tables S1 and S2”

P 8, l 155 “The participant sits 50 centimeters away from the table.” (Give distance in inches also.)

Pp 8-9 (Consider reversing the order of presentation of sections 3.2.1 and 3.2.2. Based on lines 158-160, I expected uncertainty to be discussed first. Or else reorder presentation of terms in lines 158-160 to mention expectation first.)

P 10, line 188 “Thirteen healthy students from the University of Tokyo participated in the experiment” (How is their health relevant? But give age range and gender breakdown. Also, indicate when the experiments were conducted. I’m surprised that none of the students was familiar with virtual assistants. In the US, the Amazon Echo has been around since 2015.)

L 192 (How much is 1000 Japanese Yen in US$?)

L 203 “can be found in S4 Spreadsheet”  “can be found in Spreadsheet S4”

P 11, ll 226-227 We counted the numbers of complete interactions observed during free-choice period” (How did you operationalize a ‘complete interaction’?)

P 12, l 235 “The participants could choose freely to interact with either assistant” (They had different wake words, then, one assumes. What were the wake words? Also, how were the two assistants introduced to the subjects?)

L 242 “These two categories of activities were driven by intrinsic and non-intrinsic motivation, respectively” (Explain your criteria for making these classifications. It’s not enough just to cite sources.)

P 14, ll 271-272 (A note on terminology. Instead of ‘subjective’ and ‘objective’, less value-laden and more descriptive terms would be ‘self-reported’ and ‘observed behavior’.)

P 16, l 329 (Give age range and gender breakdown of participants. Also, when did Experiment 2 take place in relation to Experiment 1?)

Ll 339-340 “Only two participants made conversation with the large-uncertainty virtual assistant; the other participants only interacted with the small-uncertainty virtual assistant.” (So this is how the two devices were differentiated – was this also the case in Experiment 1? Their roles should be clarified.)

L 344 “The number of meaningful interactions” (In what sense are they “meaningful”? A better characterization might be “complete exchanges, consisting of an initiation by the agent and a response by the assistant.”)

P 20, ll 404-407 “Non-intrinsic motivation is derived from the need to achieve an external goal, which in the context of virtual assistant interaction involves testing and making trials. Intrinsic motivation is derived from the expected enjoyment of the action itself; it is responsible for activities such as play”

(Where does seeking answers to self-generated information questions, such as “what is tomorrow’s weather?” fall according to this classification? They are not asked to test or make trials, nor to play, but rather to gain information for one’s own use.)

P 21, ll 426-430 “The pleasure perceived when the work finally makes sense is termed the “second reward” [20], as opposed to the first reward from simply receiving the information. The informative response strategy works the same way. It provides the agent with crucial information to resolve the uncertainty induced by the first few words when the assistant reports that it fails to cope.”

(I very much like the analogy to uncertainty resolution in artistic works, but the last sentence as worded does not convince me that the two situations are very similar. I think you can make a stronger case by rewording it.)

P 22, l 436 “Nevertheless, if we wish to apply such knowledge to foster intrinsically motivated interaction, it is effective to create a second reward.” (Elaborate: create a second reward by doing … ?)

L 439 “recognize users’ intentions by fitting slots.” (Explain or paraphrase this term.)

===

6. PLOS authors have the option to publish the peer review history of their article (what does this mean?). If published, this will include your full peer review and any attached files.

Reviewer #1: No

Reviewer #2: **Yes: **Susan Herring

---

## [Author Response · Author response to Decision Letter 0]

14 Dec 2020

Dear Editor and Reviewers,

Thank you for the detailed comments. We have addressed all comments in the Response to Reviewers file.

With the current revised version, we reduced abstractness by weaving in real-world examples and illustration/graphs.

To better introduce experiment variables and settings, especially the roles of the virtual assistants in terms of expectation and uncertainty, we rewrote the Introduction and Method chapters. We also added a Support Information file which presents detailed experimental procedure.

Furthermore, for the better reproducibility, we are disclosing audio files used to apply Wizard of Oz method.

We are looking forward to further discussion about the revised manuscript.

Thank you

Sincerely Yours

Chang Li

---

## [Decision Letter · Decision Letter 1]

2 Feb 2021

PONE-D-20-16053R1

Intrinsic motivation in virtual assistant interaction for fostering spontaneous interactions

PLOS ONE

Dear Dr. Li,

Thank you for submitting your manuscript to PLOS ONE. After careful consideration, we feel that it has merit but does not fully meet PLOS ONE’s publication criteria as it currently stands. Therefore, we invite you to submit a revised version of the manuscript that addresses the points raised during the review process.

We look forward to receiving your revised manuscript.

Kind regards,

Stefano Triberti, Ph.D.

Academic Editor

PLOS ONE

Reviewers' comments:

Reviewer's Responses to Questions

**Comments to the Author**

1. If the authors have adequately addressed your comments raised in a previous round of review and you feel that this manuscript is now acceptable for publication, you may indicate that here to bypass the “Comments to the Author” section, enter your conflict of interest statement in the “Confidential to Editor” section, and submit your "Accept" recommendation.

Reviewer #1: (No Response)

Reviewer #2: (No Response)

2. Is the manuscript technically sound, and do the data support the conclusions?

Reviewer #1: Yes

Reviewer #2: Yes

3. Has the statistical analysis been performed appropriately and rigorously? 

Reviewer #1: Yes

Reviewer #2: I Don't Know

4. Have the authors made all data underlying the findings in their manuscript fully available?

Reviewer #1: Yes

Reviewer #2: Yes

5. Is the manuscript presented in an intelligible fashion and written in standard English?

Reviewer #1: Yes

Reviewer #2: No

6. Review Comments to the Author

Reviewer #1: In the revised manuscript "Intrinsic motivation in virtual assistant interaction for fostering spontaneous interactions", the authors have addressed all comments improving the quality and presentation of the article. There are still few minor suggestions which would slightly improve readability and interest to the readers.

- after the first sentence of the abstract, a proper definition of user motivation should be introduced, in the context of virtual assistants, i.e., is it related to the amount of time users interact with such assistants? or is it when users use it just for fun as you state in Line 66?

- paper outline - starting Line 81: make sure you use the appropriate tense, i.e., "we first introduce" instead of "we will first introduce", "we summarize our findings ... and propose suggestions" instead of "we will summarize our findings ... and proposed suggestions"

- A reference is needed for the statement at the paragraph from Line 93.

- Line 108: ("I need to book a flight")

- After Line 118, an example of "interact for fun" with a virtual agent would be helpful at this point. It can be one of the tasks used in the study to help readers understand.

- In your analysis, several tests are used (t-tests, anova, etc.). It would be useful to add one sentence for why each test was used.

- In general, avoid using subjective language for the analysis of the results. To be specific, I would not use the phrase "unexpected findings" for the first experiment. On the contrary you could state it in a more objective way, stating that since the hypothesis was rejected, the model was revised and a verification experiment was designed.

Reviewer #2: Rereview of “Intrinsic motivation in virtual assistant interaction for fostering spontaneous interactions” for PLOS ONE

Overall: I see that you have made substantial revisions to the writing, and the result is that the presentation of the study and its background assumptions are much clearer now. The only exception is that I still struggle somewhat with all the variables in the Discussion of Figure 2. It would be helpful to make a hypothesis about uncertainty and motivation first, like you do with expectation and motivation, before adding the other variables (expectation and task difficulty) into the mix. Is that possible? Or is there a sound reason why you haven’t that?

More fundamentally, I’m also a bit concerned that since you assume expectation is affected by uncertainty/inconsistency, uncertainty and expectation are not logically independent constructs. I am curious to know how you respond to this concern.

There is now a profusion of tables and charts in the paper; it might be good to reduce their number, if possible. At least, I recommend not adding any more.

The written English is nonnative; there are numerous grammatical errors. I note that you have already promised to pay an English-language editor for the final version, though.

===

More detailed comments/suggestions follow:

Ll 32-34 “The results also suggested suppressive effects by uncertainty on intrinsic motivation, though we had not hypothesized before experiments.” (Sentence is somewhat unclear, also ungrammatical.)

Ll 51-52 “and human-robot interaction (HRI; though only those robots who interact with users as conversational agents are of our interest),” (Why mention HRI in this paper at all? It creates potential confusion, it seems to me. If the implications of your experiments extend to HRI, you could just mention that towards the end, in addressing the broader implications of the study. Or omit it entirely.)

Ll 62-63 “Unfortunately, virtual assistants are typically treated as autonomous tools by adult users” (The evaluation ‘unfortunately’ seems inappropriate here. More neutral: “However,”)

L 85 “a two-by-two experiment, which will be introduced in Chapter” (It is conventional to use the simple present tense, rather than the future tense, in describing the contents of the paper.)

L 97 “When the observation is unexpected”  “When what is observed is unexpected,”

Ll 134-135 “human’s posterior perception” (What is that? Perception “after the fact”? Define the term. The word ‘posterior’ has other meanings in English that could cause confusion. Less ambiguous terms would be Latin ‘a posteriori’ vs. ‘a priori.’ Or you could replace posterior with “after the fact.”)

L 138 Please briefly explain what the “Bayesian Estimator” is and what it does.

Ll 157-160 “Although the original model distinguished between uncertainty and noise, in that noise flattens the likelihood distribution whereas uncertainty flattens the prior distribution, we consider ‘noise’ as the uncertainty perceived by the user over the course of multiple interactions.” (What is the reasoning behind this change?)

L 196 “if not non-existing”  “if not non-existent”

Ll 207- onwards. I was following the exposition well up to this point, but I’m still struggling to wrap my head around all the variables in the Discussion of figure 2. It’s not just expectation level and uncertainty level that interact here, but also task difficulty – 3 variables, each with two values. It would be helpful to make a hypothesis about uncertainty independent of the other two variables first, like you do with expectation, before adding the other variables into the mix. … After reading a little further, I also wonder: Since you assume that expectation is affected by uncertainty/inconsistency, can uncertainty be separated from expectation with conversation agents? If not, are you really analyzing two separate constructs? On a conceptual level, I question if they are logically independent variables.)

Ll 220-221 “To sum up, we hypothesized a positive effect by expectation on motivation, and an interaction effect by expectation and uncertainty on motivation” (Is there any direct hypothesized effect of uncertainty on motivation? It seems that you get to this in the revised experimental protocol. Why not hypothesize it up front?)

L 237 “the participant could interact with the real Amazon Echo assistants” (In what language?)

L 253 “‘sculptured’” (The word you’re aiming for is “sculpted,” I think, but “constructed” might be better.)

L 314 “a Philips Hue” (needs another word. Bulb? Bridge?)

Ll 394-396 “However, Weber [et.al  et al.] [7] classified interactions which serve the purpose to ‘test the conversational agents’ understanding of language and to push it to its limits’ as ‘trolling’, which neither falls into entertainment nor utilitarian uses.” (Actually, malicious entertainment is one of the main reasons for internet trolling, according to studies that have interviewed trolls. But this usage with conversational agents seems more benign. Be aware that the word trolling normally has quite negative connotations, though. The next sentence about Friston et al. is more the perspective I think you want to adopt.)

L 452 “when a precise goal is extant”  “when a precise goal exists”

L 456 “When uncertainty is [extant  exists]”

L 461-462 “when there is no precise goal, the author argued [that] epistemic value can dominate action selection.” (The author here is Friston et al.?)

L 465 (Indicate source of the ‘original model’ in the label for Fig. 16.)

L 564 “Participants on average interacted 5.6667 times” (Reduce number of decimal places in reporting results. Your sample size does not justify this degree of precision. Two decimal places are sufficient.)

Supplementary materials:

In Table S6a, the content of the right cell of row 8 is incorrect.

Table S6B, “The virtual assistant was made to appear unable to cope with tasks 3, 7, 9 and 12, and these responses were not” (word missing – “informative”?)

As a general observation, there is now a profusion of tables and charts in the paper; it might be good to reduce their number, if possible. For example, is Table S7 essential, since its contents are a subset of Table S3? Could the relevant items just be noted where Table S7 is referred to? If you prefer to keep S7 it’s ok, but I recommend not adding any more figures/tables.

==

7. PLOS authors have the option to publish the peer review history of their article (what does this mean?). If published, this will include your full peer review and any attached files.

Reviewer #1: No

Reviewer #2: No

---

## [Author Response · Author response to Decision Letter 1]

28 Mar 2021

Dear Editor and Reviewers:

Thank you for feedback and comments on our revision. We addressed each comment and also had a proofreading team check the manuscript. In the ensuing paragraphs we cite and respond to the reviewers’ comments. Please refer to the line numbers to locate modifications/revisions. Also, please note that we have replaced “chapter” with “section”, since “chapter” is rather used in books.

Reviewer 1

Comment 1

After the first sentence of the abstract, a proper definition of user motivation should be introduced, in the context of virtual assistants, i.e., is it related to the amount of time users interact with such assistants? or is it when users use it just for fun as you state in Line 66?

Lines 22-25. We added brief definition of intrinsic and non-intrinsic motivation.

Comment 2

Paper outline - starting Line 81: make sure you use the appropriate tense, i.e., "we first introduce" instead of "we will first introduce", "we summarize our findings ... and propose suggestions" instead of "we will summarize our findings ... and proposed suggestions"

Corrected. Starting from line 80.

Comment 3

A reference is needed for the statement at the paragraph from Line 93.

Line 91. The reference here is:

Krippendorff K. The semantic turn: A new foundation for design. 1st ed. Boca Raton, FL: CRC Press; 2005.

Comment 4

Line 108: ("I need to book a flight")

Line 106. We corrected this sentence.

Comment 5

After Line 118, an example of "interact for fun" with a virtual agent would be helpful at this point. It can be one of the tasks used in the study to help readers understand.

Line 117. Tasks used in the study were primarily non-intrinsically motivating. We added an example: chatting.

Comment 6

In your analysis, several tests are used (t-tests, anova, etc.). It would be useful to add one sentence for why each test was used.

Lines 346 & 432. We added explanation for our choices. Two-way ANOVA was chosen to test interaction effect (expectation and uncertainty). T-test was chosen to investigate effects on single variables.

Comment 7

In general, avoid using subjective language for the analysis of the results. To be specific, I would not use the phrase "unexpected findings" for the first experiment. On the contrary you could state it in a more objective way, stating that since the hypothesis was rejected, the model was revised and a verification experiment was designed.

Line 442. We corrected subjective wording.

Reviewer 2

Comment 1

Ll 32-34 “The results also suggested suppressive effects by uncertainty on intrinsic motivation, though we had not hypothesized before experiments.” (Sentence is somewhat unclear, also ungrammatical.)

Line 33. We removed this statement. Instead, we clarified that the first hypothesis was not supported.

Comment 2

Ll 51-52 “and human-robot interaction (HRI; though only those robots who interact with users as conversational agents are of our interest),” (Why mention HRI in this paper at all? It creates potential confusion, it seems to me. If the implications of your experiments extend to HRI, you could just mention that towards the end, in addressing the broader implications of the study. Or omit it entirely.)

Line 58. We omitted HRI. 

Comment 3

Ll 62-63 “Unfortunately, virtual assistants are typically treated as autonomous tools by adult users” (The evaluation ‘unfortunately’ seems inappropriate here. More neutral: “However,”)

Lines 45 & 61. Thank you. We checked the document and corrected subjective languages.

Comment 4

L 85 “a two-by-two experiment, which will be introduced in Chapter” (It is conventional to use the simple present tense, rather than the future tense, in describing the contents of the paper.)

Thank you. We addressed the tense issue starting from line 84.

Comment 5

L 97 “When the observation is unexpected”  “When what is observed is unexpected,”

Line 95. Corrected.

Comment 6

Ll 134-135 “human’s posterior perception” (What is that? Perception “after the fact”? Define the term. The word ‘posterior’ has other meanings in English that could cause confusion. Less ambiguous terms would be Latin ‘a posteriori’ vs. ‘a priori.’ Or you could replace posterior with “after the fact.”)

Starting from line 135. We replaced these terms with ‘a posteriori’ and ‘a priori’.

Comment 7

L 138 Please briefly explain what the “Bayesian Estimator” is and what it does.

Line 140: We now briefly explain each term in the Bayesian estimator (Equation 1) in general sensory perception scenario, before moving to the context of assistant interaction.

Comment 8

Ll 157-160 “Although the original model distinguished between uncertainty and noise, in that noise flattens the likelihood distribution whereas uncertainty flattens the prior distribution, we consider ‘noise’ as the uncertainty perceived by the user over the course of multiple interactions.” (What is the reasoning behind this change?)

Line 162. We removed this inappropriate statement about ‘noise’, since we did not introduce ‘noise’ to our experiment. According to the original model by Yanagisawa et.al. [13], ‘noise’ is defined as the impediment to the ‘information gathering’. ‘Noise’ would be introduced, if we deliberately made one of the agents ‘speak’ through electric buzz/hum-which we did not.

Throughout the task sections, inconsistent performance induces uncertainty. Such uncertainty eventually flattens the a priori distribution, right before the free-choice period. In our experiment, uncertainty is considered the only cause of flattened a priori distribution.

Comment 9

L 196 “if not non-existing”  “if not non-existent”

Line 198. Thank you. Corrected.

Comment 10

Ll 207- onwards. I was following the exposition well up to this point, but I’m still struggling to wrap my head around all the variables in the Discussion of figure 2. It’s not just expectation level and uncertainty level that interact here, but also task difficulty – 3 variables, each with two values. It would be helpful to make a hypothesis about uncertainty independent of the other two variables first, like you do with expectation, before adding the other variables into the mix. … After reading a little further, I also wonder: Since you assume that expectation is affected by uncertainty/inconsistency, can uncertainty be separated from expectation with conversation agents? If not, are you really analyzing two separate constructs? On a conceptual level, I question if they are logically independent variables.)

Line 164. Two points should be addressed in this comment.

1. Regarding the third variable, task difficulty.

2. Regarding whether uncertainty and expectation are logically independent variables.

In response to 1: Task difficulty is not considered as a variable; it is a benchmark, which the expectation of capability is compared with. As explained in ll. 211, task difficulty is equal to least ‘capability’ required to cope with a task.

In response to 2: We need to clarify the definition of expectation. Using Fig 2, expectation is defined as the mean of probability distribution, whereas uncertainty is the variation. This definition is borrowed from the previous work of Yanagisawa et.al [13]. Therefore, expectation is not affected by uncertainty; these two are independent variables.

To dispel the confusion, consider the following structure:

a) The user intends to use the assistant to book a flight.

b) Before talking to the assistant, the user estimates whether the assistant is capable of booking a flight. (Comparing expectation of capability against the benchmark, i.e., task difficulty.)

c) The estimated outcome can be anywhere between ‘The assistant can do this’ or ‘I don’t think it can…’

The estimated outcome is indeed affected by both expectation and uncertainty. Important is, the estimated outcome and expectation of capability is two different things.

Comment 11

Ll 220-221 “To sum up, we hypothesized a positive effect by expectation on motivation, and an interaction effect by expectation and uncertainty on motivation” (Is there any direct hypothesized effect of uncertainty on motivation? It seems that you get to this in the revised experimental protocol. Why not hypothesize it up front?)

Line 222. In our research, we did not realize that uncertainty might have effects on motivation by itself until we analyzed data from the first experiment. The finding from the first experiment was essential for us to propose the revised model. Therefore, we insisted to present both experiments in the natural order.

Comment 12

L 237 “the participant could interact with the real Amazon Echo assistants” (In what language?)

Line 240. In English.

Comment 13

L 253 “‘sculptured’” (The word you’re aiming for is “sculpted,” I think, but “constructed” might be better.)

Line 255. Thank you. We now use ‘constructed’.

Comment 14

L 314 “a Philips Hue” (needs another word. Bulb? Bridge?)

Line 315. A Philips Hue LED bulb.

Comment 15

Ll 394-396 “However, Weber [et.al  et al.] [7] classified interactions which serve the purpose to ‘test the conversational agents’ understanding of language and to push it to its limits’ as ‘trolling’, which neither falls into entertainment nor utilitarian uses.” (Actually, malicious entertainment is one of the main reasons for internet trolling, according to studies that have interviewed trolls. But this usage with conversational agents seems more benign. Be aware that the word trolling normally has quite negative connotations, though. The next sentence about Friston et al. is more the perspective I think you want to adopt.)

Line 395. We removed the contents about trolling, since the ensuing taxonomy by Friston suffices. Although Weber et.al. used the word ‘trolling’ in the context of conversational agent interactions, we do realize that it is a negative expression

Comment 16

L 452 “when a precise goal is extant”  “when a precise goal exists”

Line 453. Corrected.

Comment 17

L 456 “When uncertainty is [extant  exists]”

Line 457. Corrected

Comment 18

L 461-462 “when there is no precise goal, the author argued [that] epistemic value can dominate action selection.” (The author here is Friston et al.?)

Line 462. Yes, the author is Friston et al [12].

Comment 19

L 465 (Indicate source of the ‘original model’ in the label for Fig. 16.)

Line 466. Added: reference Friston et al [12].

Comment 20

L 564 “Participants on average interacted 5.6667 times” (Reduce number of decimal places in reporting results. Your sample size does not justify this degree of precision. Two decimal places are sufficient.)

Line 545. Corrected.

Comment 21

In Table S6a, the content of the right cell of row 8 is incorrect.

Corrected.

Comment 22

Table S6B, “The virtual assistant was made to appear unable to cope with tasks 3, 7, 9 and 12, and these responses were not” (word missing – “informative”?)

Yes, we missed “informative”. Corrected.

---

## [Editor Report · Decision Letter 2]

6 Apr 2021

Intrinsic motivation in virtual assistant interaction for fostering spontaneous interactions

PONE-D-20-16053R2

Dear Dr. Li,

We’re pleased to inform you that your manuscript has been judged scientifically suitable for publication and will be formally accepted for publication once it meets all outstanding technical requirements.

Kind regards,

Stefano Triberti, Ph.D.

Academic Editor

PLOS ONE
---

## [Editor Report · Acceptance letter]

13 Apr 2021

PONE-D-20-16053R2 

Intrinsic motivation in virtual assistant interaction for fostering spontaneous interactions 

Dear Dr. Li:

I'm pleased to inform you that your manuscript has been deemed suitable for publication in PLOS ONE. Congratulations! Your manuscript is now with our production department. 

Kind regards, 

on behalf of

Dr. Stefano Triberti 

Academic Editor

PLOS ONE